# Evolutionary dynamics in gut-colonizing *Candida glabrata* during caspofungin therapy: Emergence of clinically important mutations in sphingolipid biosynthesis

Yasmine Hassoun[1], Ariel A. Aptekmann[1], Mikhail V. Keniya[1], Rosa Y. Gomez[1], Nicole Alayo[1], Giovanna Novi[1], Christopher Quinteros[1], Firat Kaya[1], Matthew Zimmerman[1], Diego H. Caceres[2,3,4], Nancy A. Chow[2], David S. Perlin[1,5,6]*, Erika Shor[1,5]*

**1** Hackensack Meridian Health Center for Discovery and Innovation, Nutley, New Jersey, United States of America, **2** Mycotic Diseases Branch, Centers for Disease Control and Prevention, Atlanta, Georgia, United States of America, **3** Center of Expertise in Mycology Radboudumc/CWZ, Nijmegen, The Netherlands, **4** Studies in Translational Microbiology and Emerging Diseases (MICROS) Research Group, School of Medicine and Health Sciences, Universidad del Rosario, Bogota, Colombia, **5** Hackensack Meridian School of Medicine, Nutley, New Jersey, United States of America, **6** Georgetown University Lombardi Comprehensive Cancer Center, Washington, D.C., United States of America

* david.perlin@hmh-cdi.org (DSP); erika.shor@hmh-cdi.org (ES)

**Data Availability Statement:** All next generation sequencing data associated with this study have been deposited at NCBI under accession number

## Abstract

Invasive fungal infections are associated with high mortality, which is exacerbated by the limited antifungal drug armamentarium and increasing antifungal drug resistance. Echinocandins are a frontline antifungal drug class targeting β-glucan synthase (GS), a fungal cell wall biosynthetic enzyme. Echinocandin resistance is generally low but increasing in species like *Candida glabrata*, an opportunistic yeast pathogen colonizing human mucosal surfaces. Mutations in GS-encoding genes (*FKS1* and *FKS2* in *C. glabrata*) are strongly associated with clinical echinocandin failure, but epidemiological studies show that other, as yet unidentified factors also influence echinocandin susceptibility. Furthermore, although the gut is known to be an important reservoir for emergence of drug-resistant strains, the evolution of resistance is not well understood. Here, we studied the evolutionary dynamics of *C. glabrata* colonizing the gut of immunocompetent mice during treatment with caspofungin, a widely-used echinocandin. Whole genome and amplicon sequencing revealed rapid genetic diversification of this *C. glabrata* population during treatment and the emergence of both drug target (*FKS2*) and non-drug target mutations, the latter predominantly in the *FEN1* gene encoding a fatty acid elongase functioning in sphingolipid biosynthesis. The *fen1* mutants displayed high fitness in the gut specifically during caspofungin treatment and contained high levels of phytosphingosine, whereas genetic depletion of phytosphingosine by deletion of *YPC1* gene hypersensitized the wild type strain to caspofungin and was epistatic to *fen1Δ*. Furthermore, high resolution imaging and mass spectrometry showed that reduced caspofungin susceptibility in *fen1Δ* cells was associated with reduced caspofungin binding to the plasma membrane. Finally, we identified several different *fen1* mutations in clinical *C. glabrata* isolates, which phenocopied the *fen1Δ* mutant, causing reduced

PRJNA1010358. The link to the dataset is https://www.ncbi.nlm.nih.gov/bioproject/1010358.

**Funding:** This work was supported by grant 5R01AI109025 from the National Institute for Allergy and Infectious Diseases to DSP and ES. The funders had no role in study design, data collection and analysis, decision to publish, or preparation of the manuscript.

**Competing interests:** DSP received an honorarium from N8 Biomedical, a mutual fund with more than $5,000 Merck stock, and an unlicensed patent for echinocandin resistance.

caspofungin susceptibility. These studies reveal new genetic and molecular determinants of clinical caspofungin susceptibility and illuminate the dynamic evolution of drug target and non-drug target mutations reducing echinocandin efficacy in patients colonized with *C. glabrata*.

## Author summary

Invasive fungal infections cause high mortality due to our limited antifungal drug armamentarium and increasing antifungal drug resistance. Echinocandins are a frontline antifungal class with increasing resistance in *Candida glabrata*, an opportunistic pathogen colonizing host mucosal surfaces. It is known that echinocandin-resistant *C. glabrata* isolates emerge from drug-sensitive gut-colonizing strains, but there are large knowledge gaps regarding how this resistance develops and whether it involves mutations in genes other than those encoding the echinocandin drug target. We studied the evolutionary dynamics in *C. glabrata* colonizing the mouse gut during treatment with caspofungin, a widely used echinocandin. In addition to well-described drug target mutations, we identified mutations in sphingolipid biosynthesis gene *FEN1* rapidly and frequently emerging in the gut-colonizing fungus during caspofungin therapy. We also identified multiple loss-of-function *fen1* mutations in clinical *C. glabrata* isolates and showed that they contribute to reduced caspofungin sensitivity of these strains. Together, this work illuminates the rich evolutionary dynamics of gut-colonizing fungi during antifungal therapy and identifies a new genetic determinant of reduced clinical caspofungin susceptibility.

## Introduction

Invasive fungal infections cause a significant health burden worldwide, with associated mortality equaling or surpassing that of malaria and TB [1–4]. Invasive *Candida* infections are associated with a mortality of >40% even with antifungal treatment [3,5,6]. The epidemiology of invasive candidiasis has been changing globally, with formerly predominant *C. albicans* decreasing in prevalence, while non-*albicans Candida* species, such as *C. glabrata* (recently reclassified as *Nakaseomyces glabratus*), *C. parapsilosis*, and *C. tropicalis* now account for ever greater fractions of invasive *Candida* infections [5,7,8]. Alarmingly, the non-*albicans Candida* species are characterized by reduced susceptibility to the azole antifungal drug class, which has been a global staple of antifungal therapy for several decades [9]. It has been proposed that the extensive use of azoles, both in medicine and in agriculture, has driven this epidemiological shift [10]. Due to this changing epidemiology, the other frontline antifungal drug class, the echinocandins, is becoming the treatment of choice for invasive candidiasis. Echinocandins have an excellent safety profile and are highly efficacious against *Candida* infections [11,12]. However, as echinocandins have been used clinically for over 20 years, resistance to this antifungal drug is also rising, most prominently in *C. glabrata* [13–18], where many cases of rapid emergence of echinocandin resistance during therapy have been documented [19–24].

Echinocandins, which are fungicidal in *Candida*, target the biosynthesis of the fungal cell wall by inhibiting the enzyme β-glucan synthase [25]. The best understood echinocandin resistance mechanism is via mutations in genes encoding β-glucan synthase (*FKS1* and *FKS2* in *C. glabrata*) that alter the structure of the enzyme, rendering it less susceptible to inhibition

[14,26,27]. While *fks* mutations have been firmly linked to *C. glabrata* echinocandin resistance leading to clinical failure, there is also evidence for a contribution of additional, as yet unidentified genetic mechanisms. For instance, in many epidemiological studies a significant proportion (up to 40%) of resistant *C. glabrata* isolates do not contain *fks* mutations [16–18,28]. Furthermore, although the mechanism of action of different echinocandins (caspofungin, micafungin, and anidulafungin) is thought to be identical, it is not uncommon for *C. glabrata* strains to show resistance to one or two but not all three of these drugs [15–18,29]. Thus, it is clear that there exist additional, as yet unknown, important determinants of susceptibility of *C. glabrata* to all echinocandins as well as to individual drugs of this class.

A number of studies have attempted to elucidate the genetic underpinnings of echinocandin resistance in *C. glabrata* by comparing whole genome sequences of resistant and susceptible clinical isolates [24,30–34]. This approach is complicated by the high genetic diversity of *C. glabrata*, where thousands or even tens of thousands of SNPs and indels can separate different clinical isolates [35]. Thus, genome comparisons typically identify many putative genetic changes regulating echinocandin sensitivity, but validating the role of each individual SNP is extremely labor-intensive and usually not carried out, nor are the mechanisms by which these SNPs may alter echinocandin sensitivity elucidated. In another approach to identify the missing genetic basis for echinocandin resistance, *in vitro* evolution has been used to show that *C. glabrata* passaged in rich laboratory medium containing increasing concentrations of anidulafungin acquires mutations in *FKS1* and *FKS2* as well as mutations in ergosterol biosynthesis gene *ERG3*, implicating the latter in echinocandin resistance via as yet unknown mechanisms [36]. However, it is not yet clear how relevant this narrow mutational signature of resistance is to the mutations driving the evolution of echinocandin resistance in the context of the host. Importantly, the gastrointestinal (GI) tract and the abdominal cavity were identified as key host reservoirs in which *C. glabrata* drug resistance emerges [37–39] and *in vitro* growth in rich medium may not recapitulate the evolutionary dynamics of *C. glabrata* existing within these host niches.

In this study, we examined the evolution of resistance to the echinocandin caspofungin in *C. glabrata* colonizing the mouse gut. We found that both *fks* and non-*fks* mutations arose in GI-colonizing *C. glabrata* during treatment with caspofungin. The majority of non-*fks* mutations were in the *FEN1* gene, which encodes a fatty acid synthase and functions in sphingolipid biosynthesis. Further analysis demonstrated that *fen1* mutants have a fitness advantage in the gut specifically during caspofungin treatment, that these mutants accumulate phytosphingosine (PHS, a sphingolipid biosynthesis intermediate), and that genetic depletion of PHS by deletion of alkaline ceramidase *YPC1* sensitizes both the wild type strain and the *fen1* mutant to caspofungin. Interestingly, in-depth analysis of the emergence and abundance of both drug target (*fks*) and non-drug target (*fen1*) mutations during caspofungin treatment by amplicon sequencing revealed a rapid emergence and diversification of caspofungin-adapted strains in the GI tract. Importantly, we also identified three different point mutations in *FEN1* in *C. glabrata* clinical isolates and showed that all of these clinical *fen1* mutations cause loss of protein function and reduced susceptibility to caspofungin. In summary, our study uncovered mutations in *FEN1* as a novel genetic mechanism reducing the susceptibility of *C. glabrata* to the widely used echinocandin caspofungin in the clinical setting and showed that this effect is due to an increase in the membrane level of PHS. Beyond its clinical implications, our work also sheds light on the complex evolutionary dynamics of *C. glabrata* within the host GI reservoir, comprising both drug target and non-target mutations, which collectively undermine the effectiveness of antifungal treatment in patients colonized with *C. glabrata*.

## Results

### Non-drug target mutations arise in *C. glabrata* colonizing the gut during caspofungin treatment

In a previous study, which focused on the role of cell wall integrity pathways in echinocandin tolerance, we examined the evolution of echinocandin resistance during caspofungin (CSF) treatment in the mouse gut *C. glabrata* colonization model [40]. Approximately half of the evolved *C. glabrata* strains showed increased MICs for both CSF and micafungin and had *FKS2* hot-spot mutations, but the rest of the strains showed increased MICs for CSF only and lacked mutations in either *FKS1* or *FKS2* [40]. We also observed the development of such non-*fks* mutants in the mouse gut during CSF treatment of mice colonized with *C. glabrata* strains lacking error-prone DNA polymerases Rev1 and Rev3. To identify the causes of reduced CSF susceptibility in these strains, we sequenced the genomes of seven non-*fks* mutants with increased CSF MICs, as well as four strains with *FKS2* hot-spot mutations that had also evolved in the mouse gut during CSF treatment (Fig 1A and S1 Table). Interestingly, we found that six out of 11 sequenced strains–five strains without *fks2* mutations and one strain with an *fks2* mutation–contained mutations in the *FEN1* gene (*ELO2* in *Saccharomyces cerevisiae*) (Fig 1A). *FEN1* encodes a fatty acid elongase that functions in sphingolipid biosynthesis [41,42] (Fig 1B). This finding was consistent with previous reports that mutations in genes of the sphingolipid biosynthesis pathway, including *FEN1*, reduce *C. glabrata* and *C. albicans* sensitivity to CSF [43,44]. Also consistent with those reports, the *fen1* mutant strains had increased susceptibility to micafungin (MCF), as well as increased or unaltered susceptibility to anidulafungin (ANF) (Fig 1A).

All gut-evolved strains also contained multiple other coding SNPs relative to the parental strain DPL1021 (ATCC90030) and each other, suggesting that these SNPs had arisen during evolution in the mouse gut (S1 Table). Among genes previously implicated in echinocandin resistance, GI-63 carried a mutation in the *ERG3* gene, which encodes an enzyme involved in the biosynthesis of ergosterol [45]. Mutations in *ERG3* gene can lead to alterations in sterol composition and confer resistance to azoles, as well as echinocandins [45–47] and have also been reported to arise during *in vitro* evolution of echinocandin resistance [36].

**A**

| Strain | Genotype | FEN1 | FKS2 | CSF MIC | MCF MIC | ANF MIC |
|--------|----------|------|------|---------|---------|---------|
| DPL1021 | WT | – | – | 0.5 | 0.06 | 0.06 |
| GI-52 | WT | – | Asp666Gly | 1 | 0.06 | 0.125 |
| GI-53 | WT | – | Arg1378Cys | 2 | 0.06 | 0.06 |
| GI-56 | *yps1Δ* | Tyr81Ser | – | 2 | 0.015 | 0.03 |
| GI-63 | *slt2Δ* | Gln125* | – | ng | ng | ng |
| GI-113 | WT | – | – | 2 | 0.015 | 0.03 |
| GI-240 | WT | Trp145* | – | 2 | 0.015 | 0.06 |
| GI-250 | *rev1Δ* | Cys192Arg | – | 2 | 0.015 | 0.06 |
| GI-297 | WT | Tyr81His | Leu664Val | 2 | 0.06 | 0.125 |
| GI-299 | WT | Ala259fs | – | 2 | 0.015 | 0.06 |
| GI-332 | *rev3Δ* | – | – | 2 | 0.015 | 0.03 |
| GI-YR01 | WT | – | Ser663Pro | 32 | 4 | n/a |

(gut-evolved during CSF treatment)

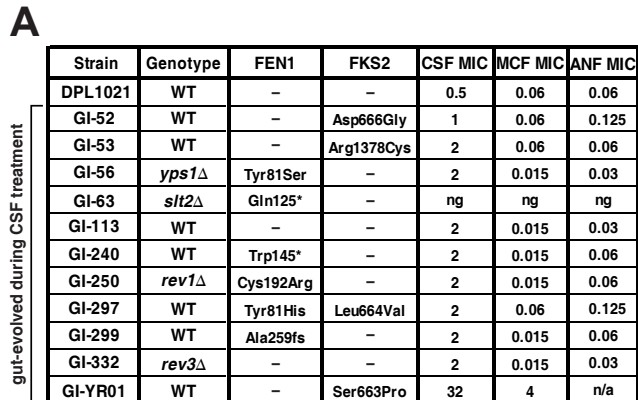

**B**

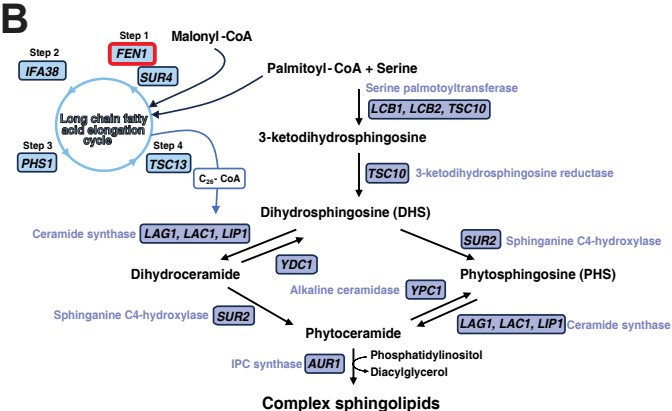

**Fig 1. Mutations in fatty acid elongase Fen1, which functions in sphingolipid biosynthesis, frequently evolved in gut-colonizing *C. glabrata* during caspofungin treatment. A.** Genetic and echinocandin susceptibility information for 11 gut-evolved mutants. All parental strains used to colonize the mice belonged to the DPL1021 (ATCC90030) background. Some of the parental strains carried the gene deletions shown in the "Genotype" column; "WT" indicates the absence of such gene deletions. These gene deletions had been made as part of our investigation of the roles of the cell wall integrity pathway (*slt2Δ* and *yps1Δ* [40]) and error-prone DNA replication (*rev1Δ* and *rev3Δ*) in the evolution of CSF-resistant mutants in gut-colonizing *C. glabrata*. **B.** Diagram of the yeast sphingolipid biosynthesis pathway. *FEN1*, which encodes a fatty acid elongase, is outlined in red. * = stop codon; CSF = caspofungin; MCF = micafungin; ANF = anidulafungin.

## Deletion or mutation of *FEN1* reduces CSF susceptibility in *C. glabrata* colonizing the gut but not *C. glabrata* infecting the kidney

To directly analyze the role of *FEN1* during host colonization and infection, we deleted the *FEN1* ORF and colonized mice with *fen1Δ* and *fen1-W145** (GI-240, Fig 1A). We found that daily CSF treatment (20 mg/kg) resulted in a stronger reduction in fungal burdens in mice colonized with the wild type strain than in mice colonized with the *fen1Δ* or *fen1-W145** mutants (Fig 2A). We also directly compared *in vivo* fitness of the wild type strain with that of a *fen1* mutant by colonizing the mouse gut with a mix of both strains and following each strain's abundance over seven days, with or without daily CSF treatment. We found that whereas in the absence of CSF treatment, the wild type strain was significantly more fit than either *fen1Δ* or *fen1-W145** strains, in the presence of CSF treatment the two *fen1* mutants were more fit, fully taking over the gut population by day 5 (Fig 2B). This result was consistent with our initial observation that *fen1* mutants were selected in the gut during CSF treatment and confirmed that loss of Fen1 function improved *C. glabrata* fitness in the presence of CSF.

We also analyzed the effect of *fen1* mutation on *C. glabrata* CSF susceptibility in the context of a bloodstream infection. Mice were immunosuppressed and infected via the retro-orbital route with either wild type, *fen1Δ*, or *fks2-S663P* mutant strains, the latter of which is a prominent cause of clinical echinocandin resistance and served as a control. Kidney fungal burdens were assessed as a measure of CSF efficacy as described before [48]. We used a range of CSF doses (1.25, 2.5, and 5 mg/kg), the highest of which corresponds to the humanized dose. Consistent with the documented resistance of the *fks2-S663P* mutant [26], we found that its burden in the kidney was not reduced as much as those of the wild type strain at all three concentrations, with the difference being statistically significant at 5 mg/kg (Fig 2C). In contrast, there was no difference in kidney burdens between the *fen1Δ* and the wild type strain at any of the caspofungin concentrations (Fig 2C), showing that in the context of bloodstream infection, unlike during gut colonization, the *fen1Δ* mutant remained susceptible to CSF.

## Deletion of *FEN1* results in reduced caspofungin association with the fungal plasma membrane

Sphingolipid biosynthesis has been previously implicated in regulating fungal susceptibility *in vitro* to CSF but not to other echinocandins [43,44]. However, the molecular basis for the reduced CSF susceptibility of sphingolipid mutants, such as *fen1Δ*, is not known. It has been previously proposed that the fatty acid-like tail of CSF (Fig 3A), which is not present in other echinocandins, may modulate its interaction with the plasma membrane and membrane-bound glucan synthase [43]. To test this hypothesis, we first analyzed the CSF-plasma membrane interaction using fluorescent CSF. This compound (custom synthesized and purified at WuXi Biologicals, China) was obtained by reacting a free amine group on CSF with Alexa Fluor 647 (AF647) modified with an amine-reactive NHS ester (Fig 3A). The WT and *fen1Δ* strains were incubated with CSF-AF647 and analyzed by high resolution imaging (ONi Nanoimager). The imaging showed that CSF-AF647 showed a strong association with the cellular periphery in the WT cells (Fig 3B), consistent with its association with the plasma membrane. Importantly, this association was fully dependent on the CSF moiety of the molecule, as AF647 alone (in which the amino-reactive ester had been quenched with TRIS) did not bind to *C. glabrata* cells (S1C Fig). Interestingly, the binding of CSF-AF647 to the fungal cell surface was significantly reduced in the *fen1Δ* cells, and this reduction observed when *FEN1* was deleted in strains CBS138 (Fig 3B) and ATCC90030 (S1A Fig), showing that it was independent of *C. glabrata* strain background. Thus, these observations suggested that a key difference between WT and *fen1Δ* cells is a decreased association between CSF and the plasma membrane.

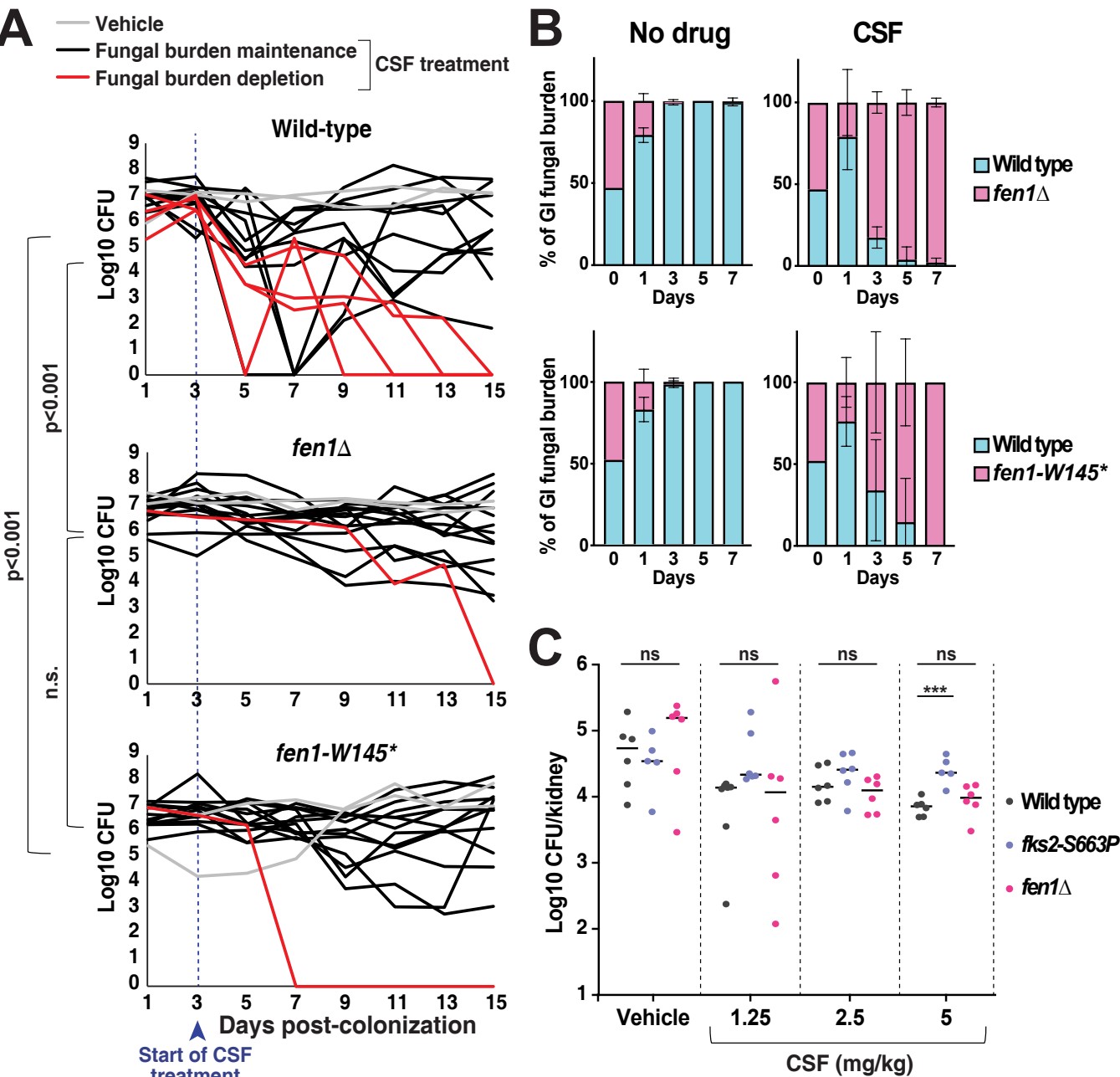

**Fig 2. Deletion or mutation of *FEN1* improves fitness of gut-colonizing *C. glabrata* during caspofungin (CSF) treatment. A.** Gut-colonizing *fen1* mutants survive during CSF therapy better than the isogenic wild type strain. The mice were colonized with the indicated *C. glabrata* strains and treated daily with CSF (20 mg/kg) starting at day 3 post-colonization. Fecal samples were obtained every other day and plated to obtain GI fungal CFU counts. Multivariate Analysis of Variance (MANOVA) was used to assess the relationship between treatments over time on gut microbiota with a significant threshold set at 0.05. Tukey's Post Hoc analysis was performed to look for pairwise significance. "Fungal burden depletion" refers to *C. glabrata* gut levels decreasing to undetectable levels during CSF treatment and not rebounding during the timeframe of the experiment. **B.** In direct competition experiments, gut-colonizing *fen1* mutants outcompete the wild type strain specifically during CSF treatment. The mice were colonized with an equal mix of two strains: wild type and an isogenic *fen1* mutant (*fen1Δ* or *fen1-W145*\*). Starting at day 1 post-colonization, the mice were treated daily with CSF (20 mg/kg). Fecal samples were plated every other day, and the resulting colonies were analyzed for their *FEN1* status (wild type or mutant). To identify *fen1Δ* colonies, the plates were replica-plated on hygromycin-containing medium. To identify *fen1-W145*\* colonies, a specific molecular beacon was designed and used in colony PCR (see Materials and Methods). **C.** In a model of systemic infection, the *fen1Δ* mutant was as sensitive to CSF treatment as the wild type strain. Neutropenic mice were infected via the retro-orbital route with the indicated strains of *C. glabrata*, and treatment with CSF (1.25, 2.5 or 5 mg/kg) or vehicle (PBS) was initiated at day 1 post-infection. The kidneys were harvested on day 3, homogenized and plated on YPD for CFU counts. Each data point corresponds to a single mouse. \*\*\* p < 0.001 (ordinary one-way ANOVA).

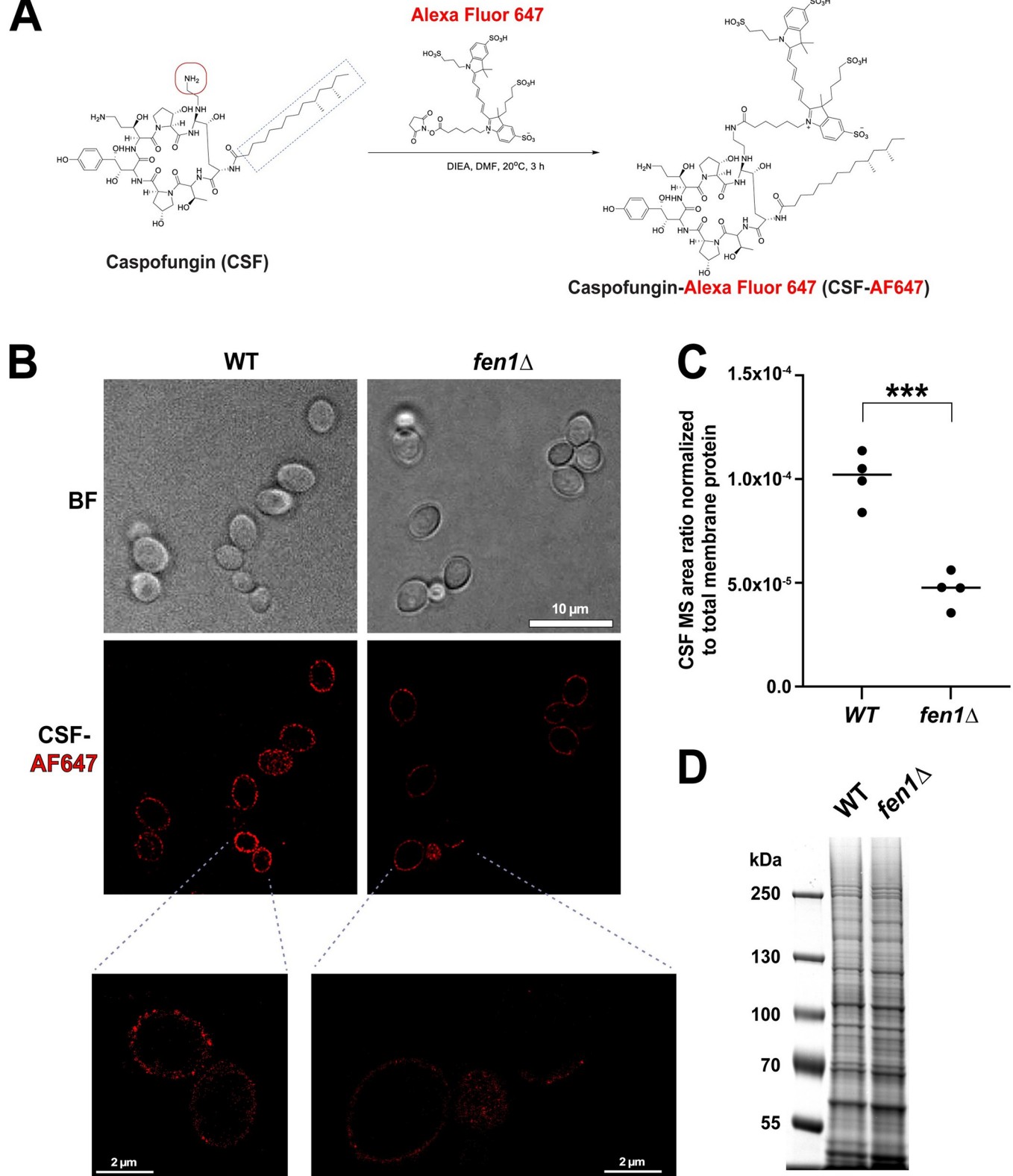

**Fig 3. Deletion of *FEN1* reduces caspofungin (CSF) binding to *C. glabrata* cells. A.** Alexa Fluor 647 was conjugated to the indicated primary amine group of CSF (red circle). The synthesis was carried out at WuXi Biologics (China). The product was confirmed by nuclear magnetic resonance and high-resolution mass spectrometry and determined to be 98.67% pure. CSF's fatty acid-like tail is indicated by the dashed blue box. **B.** High-resolution imaging showed that

CSF-AF647 strongly associated with the fungal cell surface in WT cells, but that this association was greatly reduced in the *fen1Δ* mutant. **C.** CSF interaction with cellular membranes was significantly reduced in the *fen1Δ* mutant relative to the WT strain. Cells were incubated with CSF, disrupted, and crude membranes were extracted and analyzed by LC-MS/MS. The detected CSF levels were internally normalized to each sample's protein levels. Each data point corresponds to a single biological replicate. **D.** Protein composition of the crude membrane preparations (Coomassie-stained 8% TRIS-Glycine SDS PAGE) was unaltered by *fen1Δ*. BF = bright-field; AF = Alexa Fluor. *** p<0.001 (two-tailed unpaired T-test).

Unfortunately, the addition of AF647 significantly reduced the antifungal activity of CSF-AF647 relative to CSF: the MIC of CSF-AF647 was ~0.85% of that of CSF in both WT and *fen1Δ* strains (S1B Fig). Furthermore, our mass spectrometry (MS) analysis of the CSF-AF647 preparation showed that it contained 0.6% unlabeled CSF, which may have accounted for some of the observed antifungal activity of CSF-AF647. Thus, the reduced association of CSF with plasma membrane in the *fen1Δ* mutant needed to be verified using a different approach not reliant on the CSF-AF647 conjugate. To this end, we used MS to directly measure the binding of unlabeled CSF to cellular membranes in WT and *fen1Δ* strains. *C. glabrata* cells were incubated with 0.25 µg/ml CSF for 20 minutes, followed by immediate cell harvesting and membrane extraction. 10% the membrane preparation was saved for protein quantification, and the rest was analyzed by LC-MS/MS. The detected CSF levels were internally normalized to protein levels in the same sample. Consistent with the imaging data (Fig 3B), we observed a 50% reduction in CSF association with the membranes derived from the *fen1Δ* mutant relative to the membranes derived from the WT strain (Fig 3C). On the other hand, the membrane protein band patterns of the WT and *fen1Δ* strains were very similar (Fig 3D), consistent with the conclusion that the reduced CSF association in the *fen1Δ* mutant was likely due to its altered membrane lipid composition. In summary, both imaging and MS-based analyses have shown that the interaction of CSF with the plasma membrane is significantly reduced in the *fen1Δ* mutant, potentially explaining the reduced sensitivity of this mutant to CSF.

## Manipulating phytosphingosine metabolism can sensitize *C. glabrata* to CSF

Previous studies have shown that in *C. albicans* some sphingolipid biosynthesis mutants accumulated phytosphingosine (PHS), a sphingolipid biosynthesis intermediate [44], and that increased PHS can render GS less susceptible to inhibition by echinocandins [49,50]. Thus, we performed a sphingolipid-focused lipidomic analysis of two *fen1* mutants (*fen1Δ* and GI-299) and one non-*fen1* gut-evolved strain (GI-113) both in the absence and presence of 1 µg/ml CSF. The results showed that all three gut evolved mutants (both *fen1* and non-*fen1*) had elevated levels of PHS, which were increased further during CSF exposure (Fig 4A and S2 Table). To further probe the importance of PHS accumulation in regulating CSF susceptibility, we deleted *YPC1*, the gene encoding alkaline ceramidase, which in *S. cerevisiae* catalyzes the conversion of phytoceramide to PHS [51,52] (Fig 1B). Interestingly, we observed that *ypc1Δ* had the opposite effect of *fen1Δ*, hypersensitizing *C. glabrata* to CSF, which was evident on agar plates containing CSF (Fig 4B). Furthermore, *ypc1Δ* was epistatic to *fen1Δ*, as the double mutant was as sensitive as the *ypc1Δ* alone (Fig 4B). Consistent with these results, *ypc1Δ* also restored CSF-AF647 association with the fungal cells in the *fen1Δ* mutant (Fig 4C). We also attempted to delete *YPC1* in gut-evolved strain GI-113, but despite multiple attempts, no deletants were recovered, suggesting a possible synthetic lethality with one of the mutations in that strain (S1 Table). Together, these results show that accumulation of PHS is associated with reduced CSF sensitivity and reduced CSF binding to the plasma membrane, whereas genetic depletion of PHS, conversely, sensitizes *C. glabrata* to CSF and enhances CSF association with the plasma membrane.

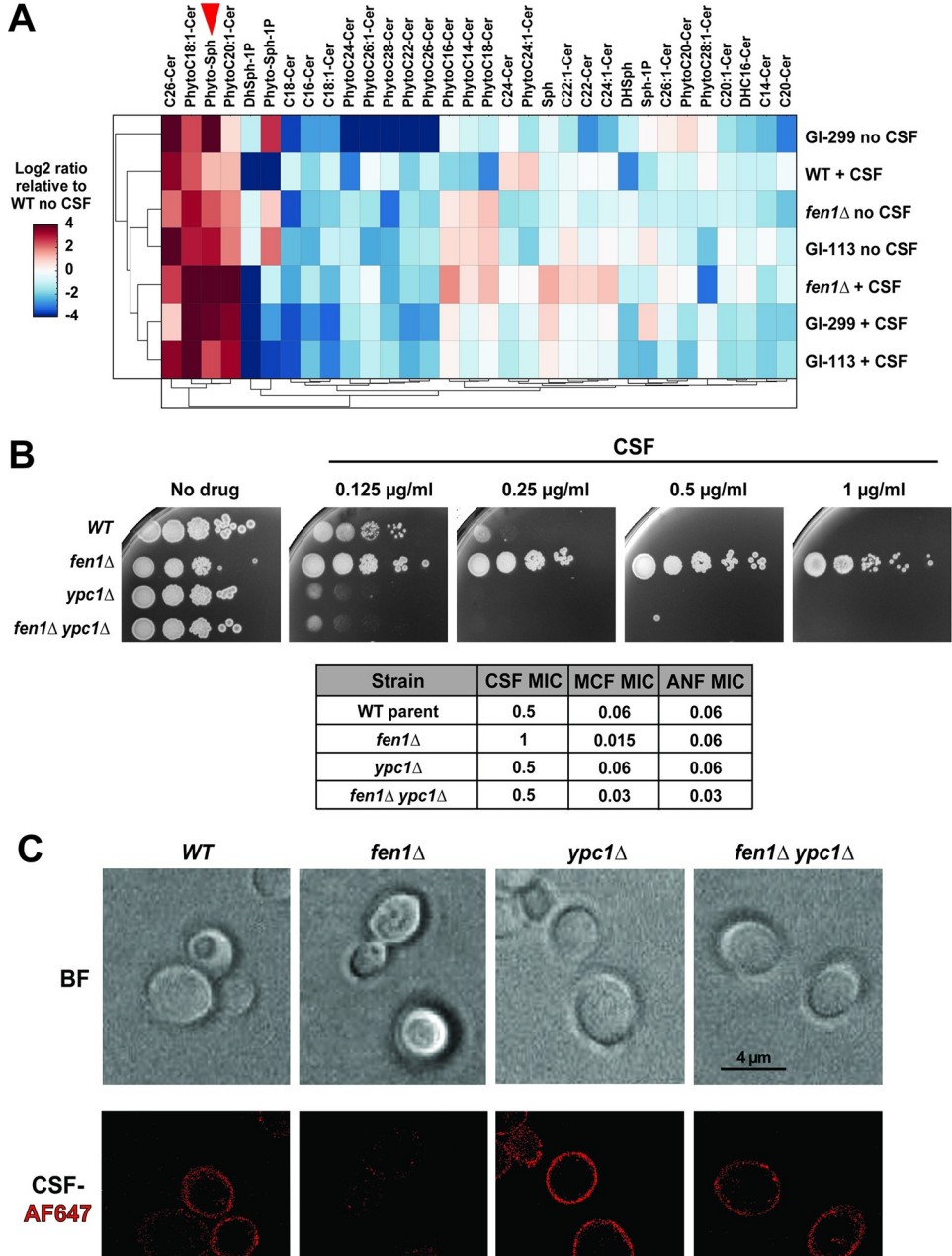

**Fig 4. Manipulating phytosphingosine levels can sensitize *C. glabrata* to caspofungin (CSF). A**. Lipidomic analysis showed that phytosphingosine (PHS, red arrowhead) levels are increased in gut-evolved CSF-adapted *C. glabrata* mutants. The indicated strains were either left untreated (No CSF) or exposed to CSF (0.5 µg/ml) for one hour. The samples were analyzed at the Lipidomics Shared Resource at the Medical University of South Carolina. All samples were normalized against inorganic phosphate and the untreated WT control. **B.** Deletion of alkaline ceramidase *YPC1*, which breaks down phytoceramide to produce PHS [51, 52], sensitized wild type and *fen1Δ C. glabrata* to CSF. This sensitization was evident when *C. glabrata* was grown on solid CSF-containing YPD medium (0.125 µg/ml and 0.25 µg/ml CSF) but not in liquid RPMI, where *ypc1Δ* did not alter the CSF MIC. Both growth on solid medium and the MIC measurement showed that *ypc1Δ* was epistatic to *fen1Δ*, sensitizing it to CSF. MCF = micafungin; ANF = anidulafungin **C.** Deletion of *YPC1* also restored CSF binding to the fungal cell surface in the *fen1Δ* strain. BF = bright-field; AF = Alexa Fluor.

### Evolutionary dynamics of *fen1* and *fks* mutations in gut-colonizing *C. glabrata* during CSF treatment

To check whether the results obtained for the strains chosen for whole genome sequencing (Fig 1A) were representative of general *C. glabrata* evolution in the gut (i.e., that *fen1* mutations arose frequently in this population during caspofungin treatment), we used yeast samples obtained and stocked during previous gut colonization experiments (Fig 5A). Because this analysis was done *post hoc*, it was done on a limited number of colonies: on those occasions, two colonies per mouse per day had been cultured, *FKS1* and *FKS2* hot spots sequenced, and frozen stocks made. We now sequenced *FEN1* as well (Fig 5A). This analysis confirmed that a variety of *fen1* mutations had arisen in the different mice, including two different *fen1* mutations co-existing in mouse 4 on day 7. Interestingly, in mice 1, 4, and 5 where both *fen1* and *fks2* mutations were observed, the *fen1* mutations had appeared earlier than *fks2* mutations (Fig 5A; no *fks1* mutations had been detected). In mice 2, 3 and 6 only *fen1* mutations were detected, and these also appeared early–on days 7 and 9, i.e., 4 and 6 days after the initiation of CSF treatment. These results suggested that the gut-colonizing population of *C. glabrata* during CSF treatment is highly dynamic and diverse, rapidly acquiring both drug target and non-drug target mutations that improve fitness in the presence of CSF.

To gain an even deeper insight into the timing and spread of *FKS* and *FEN1* mutations in *C. glabrata* colonizing the gut during CSF treatment, we used amplicon sequencing (Fig 5B). Four mice were colonized with *C. glabrata* and treated with CSF as in our previous studies [37, 40] (S2A Fig). Feces were collected on the indicated days and plated on YPD agar plates supplemented with antibiotics, as before, but instead of picking individual yeast colonies, ~1000 colonies were pooled for DNA isolation, followed by amplicon PCR and next generation sequencing. The resulting reads were mapped to the amplicon sequences to identify SNPs appearing in the gut-colonizing *C. glabrata* (Fig 5B). First, we observed that the original

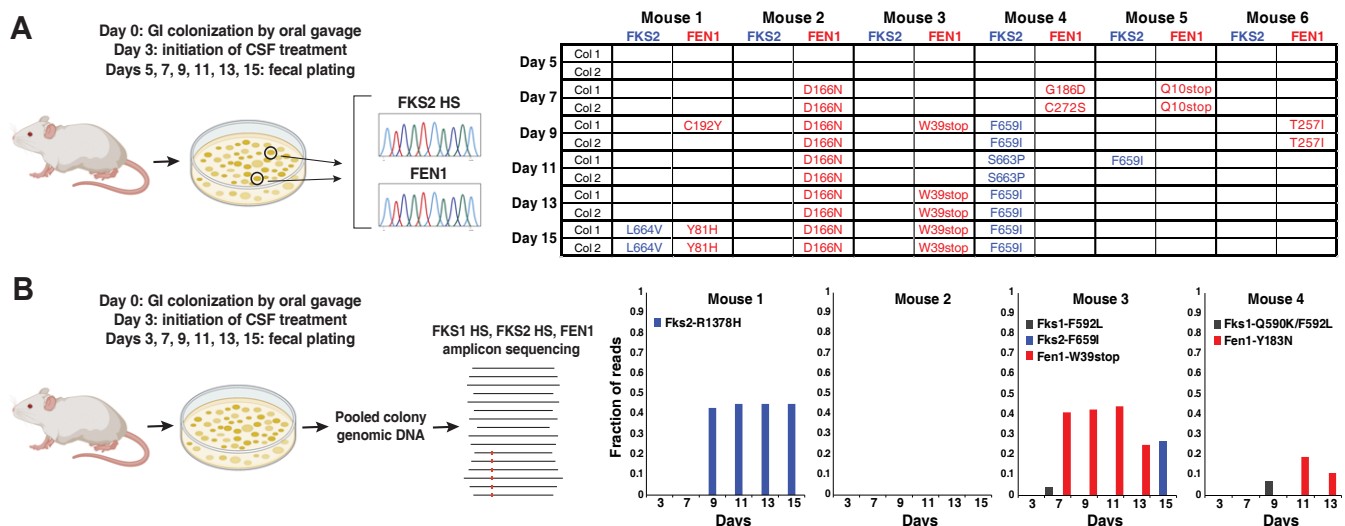

**Fig 5. Dynamics of drug target (*fks*) and non-drug target (*fen1*) mutations arising in gut-colonizing *C. glabrata* during caspofungin (CSF) treatment. A.** Sanger sequencing of *FKS1* and *FKS2* hot spots and *FEN1* ORF in individual colonies (two per mouse per day) derived from gut-colonizing *C. glabrata* reveals the wide variety of *fen1* mutations, some detected as early as day 7 (i.e., 4 days after initiation of CSF treatment). No *FKS1* hot spot mutations were found. **B.** Amplicon sequencing of *FKS1* and *FKS2* hot spots and *FEN1* ORF revealed the time of appearance and relative abundance of *fks1*, *fks2*, and *fen1* mutant alleles in gut-colonizing *C. glabrata* during CSF treatment. Fraction of reads containing the indicated allele relative to all reads containing that position was used as a proxy for the relative abundance of the mutant allele in the gut-colonizing *C. glabrata* population. Parts of Panels A and B of this figure were made using Biorender.com (lab license).

inoculum contained several SNPs in *FKS1* HS1 present at low frequencies (S2B Fig), suggesting that these mutations arose in the *C. glabrata* culture used for mouse oral gavage. These mutations tended to persist in the mouse gut at similar frequencies throughout the two-week CSF treatment, indicating that they did not affect fitness under these conditions (S2B Fig). In contrast, mutations in *FEN1* and *FKS2* appeared and rapidly increased in abundance in mice 1, 3 and 4 (Fig 5B), indicating that they increased fitness. The earliest *fen1* mutation (W39*), in mouse 3, was detected on day 7, whereas the earliest *fks2* mutation (R1378H), in mouse 1, was detected on day 9 (and in this mouse no *fen1* mutation was ever detected). Interestingly, although both of these mutations arose relatively early during the treatment, neither of them swept the population but remained at a <50% abundance (Fig 5B), suggesting that they co-existed with other fitness-promoting mutations that had arisen in other cells. Furthermore, the *fen1*-W39* mutation in mouse 3 started decreasing in abundance by day 13 and was undetected on day 15, at which point the *fks2*-F659I mutation, which is associated with clinical echinocandin resistance [53], had appeared in the same population (Fig 5B). Finally, although a few *FKS1* HS mutations had appeared in mice 3 and 4, none of them significantly increased in abundance or were detected on multiple days, indicating that they had marginal effects on fitness during CSF treatment. Thus, this experiment confirmed that GI-colonizing *C. glabrata* was highly genetically dynamic during CSF treatment, with caspofungin-adapted mutations promoting fitness arising, increasing in relative abundance, and co-existing with other such mutations.

## Clinical *C. glabrata* isolates contain loss-of-function *fen1* mutations that cause reduced susceptibility to CSF

To investigate whether *fen1* mutations also occur in *C. glabrata* isolates infecting humans, we used two approaches: mining publicly available genomes and sequencing *FEN1* in strains from our lab collection. We used two sources of clinical isolate genome data: those in Biswas et al. (52 strains) [30] and those collected by the Centers for Disease Control (CDC) as part of the Emerging Infections Program (EIP) for candidemia surveillance and made publicly available on NCBI [67]. Because strains of different sequence types (ST) are expected to contain ST-specific polymorphisms in *FEN1*, we also determined the ST of each strain, either from the whole genome sequence or by Sanger sequencing using the standard *C. glabrata* multi-locus sequence typing scheme [54]. These analyses identified three clinical *C. glabrata* isolates carrying unique mutations in *FEN1* that were not ST-associated polymorphisms: G143V in strain WM_18.26 [30], Y298* in strain DPL239 (Perlin lab collection), and G52A in strain CAS14-5989 (CDC collection) (Fig 6A and S3 and S4 Tables). WM_18.26 also had the *fks2*-S663P mutation and a high reported CSF MIC (16 μg/ml), while CAS14-5989 also contained the *fks1*-S595T mutation but a low CSF MIC (0.03 μg/ml, according to CDC records) (Fig 6A). DPL239 with *fen1*-Y298* contained wild type *FKS1* and *FKS2* sequences but an elevated CSF MIC (1 μg/ml). The combination of *fen1*-G143V with *fks2*-S663P in the genome of a differently named clinical isolate (CMRL4) has previously been reported [34]. Because both WM_18.26 and CMRL4 were originally reported by the same group within one year of each other [30,55], it is likely that they are either the same strain or two closely related strains. Interestingly, the CMRL4 strain was obtained from a patient who had had 30 days of CSF treatment [55], consistent with emergence of the *fen1*-G143V mutation during CSF therapy. No history of antifungal treatment was available for strains DPL239 and CAS14-5989, but because the latter also contained an *fks1* hot-spot mutation, it may be hypothesized that the *fen1* and *fks1* mutations may have arisen under a similar environmental pressure.

Although the *fen1*-G143V mutation was previously reported, it was not known whether it affected Fen1 function, and the same was true for the newly discovered *fen1*-G52A mutation.

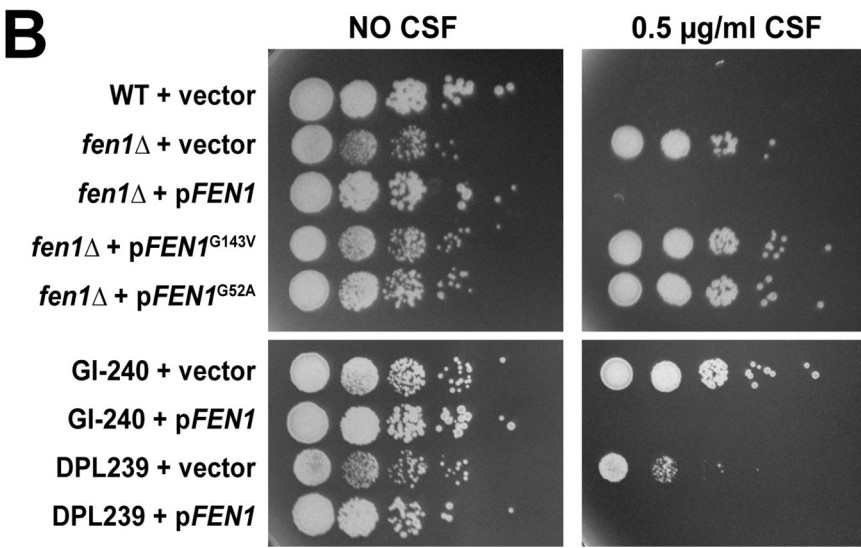

## A    *FEN1* mutations in human clinical isolates

| STRAIN NAME | STRAIN ORIGIN | FEN1 mutation | FKS mutation | CSF MIC (μg/ml) | MCF MIC (μg/ml) | ANF MIC (μg/ml) |
|---|---|---|---|---|---|---|
| WM_18.26 | Biswas 2017,2018 | G143V | Fks2-S663P | 16 | 1 | 1 |
| DPL239 | Perlin lab collection | Y298* | – | 2 | 0.03 | 0.06 |
| CAS14-5989 | CDC | G52A | Fks1-S595P | 0.03 | 0.008 | 0.06 |

**Fig 6. Clinical *C. glabrata* mutations in *FEN1* result in loss of protein function and reduced susceptibility to caspofungin (CSF). A.** Analysis of publicly available *C. glabrata* genomes and Sanger sequencing of *FEN1* in the Perlin lab strain collection identified three unique (non-ST-associated) *fen1* point mutations in clinical strains. Two of the strains also contained mutations in either *FKS1* or *FKS2*. The MIC values for CSF and micafungin (MCF) for WM_18.26 were obtained from [30], for DPL239 from our measurements, and for CAS14-5989 from CDC records. **B.** All three identified clinical *fen1* mutations phenocopy *fen1Δ* and reduce *C. glabrata* CSF sensitivity. Strains containing wild type *FEN1* do not grow in the presence of CSF, whereas strains lacking either functional chromosomal or plasmid *FEN1* grow in the presence of the drug. When introduced into *C. glabrata* on a low-copy plasmid, neither *fen1*^G143V nor *fen1*^G52A were able to complement *fen1Δ* CSF-resistant phenotype. Conversely, the CSF-resistant phenotype of clinical and gut-evolved strains carrying chromosomal *fen1* mutations was complemented by WT *FEN1* introduced on a low-copy plasmid.

Because strains WM_18.26 and CAS14-5989 were not readily available, we introduced the G143V and G52A mutations, separately, into *FEN1* carried on a low copy plasmid and transformed these plasmids into the *fen1Δ* strain. Unlike the plasmid carrying wild type *FEN1*, which fully complemented the loss of Fen1 by restoring wild type CSF sensitivity, the plasmids carrying *fen1*-G143V or *fen1*-G52A behaved like the empty vector, strongly suggesting that the *fen1*-G143V mutation caused a loss of Fen1 function (Fig 6B). To determine whether the reduced CSF sensitivity of strain DPL239 was due to the *fen1*-Y298* mutation, we transformed it either with the *FEN1* plasmid or an empty vector. Whereas the empty vector had no effect, the *FEN1* plasmid sensitized the strain to CSF (Fig 6B), strongly suggesting that the *fen1*-Y298* mutation contributes to the reduced CSF susceptibility of this clinical *C. glabrata* isolate. The same result was obtained with gut-evolved strain GI-240 (Fig 6B), confirming that it contained a loss-of-function *fen1* mutation (Fig 1A). Together, these experiments showed that loss-of-function *fen1* mutations occur in clinical *C. glabrata* strains and contribute to the strains' reduced susceptibility to CSF.

## Discussion

In this study, we investigated the evolution of resistance to the widely used echinocandin class drug caspofungin in *C. glabrata* colonizing the GI tract of immunocompetent mice. We found that, in addition to the well-described mutations in echinocandin target β-glucan synthase, other mutations, predominantly in the *FEN1* gene involved in sphingolipid biosynthesis, arose rapidly after the start of caspofungin treatment and increased in abundance in the gut due to their strong fitness advantage over the wild-type strain in the presence of the drug. Our lipidomic, genetic, high-resolution imaging, and MS analyses revealed that the reduced caspofungin sensitivity of *fen1* mutant strains is due to elevated intracellular levels of phytosphingosine (PHS), a sphingolipid biosynthesis intermediate, and is associated with reduced caspofungin binding to *C. glabrata* plasma membrane. Importantly, we also identified several different *fen1* mutations in clinical *C. glabrata* isolates, and our analyses of these mutations showed that they result in loss of protein function and reduced susceptibility to caspofungin. Together, these results identify a new genetic determinant of clinical caspofungin susceptibility in *C. glabrata* and illuminate the rich dynamics of evolutionary trajectories in gut-colonizing *C. glabrata* during caspofungin treatment.

Our study illuminates the rapid and complex evolutionary dynamics taking place in the gut *C. glabrata* reservoir during echinocandin therapy. Both Sanger sequencing of individual colonies and amplicon sequencing of pooled colonies have revealed that mutations in *FEN1* emerge and rapidly increase in relative abundance as early as 4 days after the initiation of caspofungin treatment. These results also provide evidence of mixed mutant populations of gut-commensal *C. glabrata* during caspofungin treatment, where different sub-populations carry different mutations in *FEN1*, *FKS2*, and other genes affecting caspofungin sensitivity, and where the relative abundance of these mutant sub-populations fluctuates according to their relative fitness in the presence of caspofungin. Indeed, our amplicon sequencing analysis demonstrated that *fen1-W39** and *fks2-R1378H* alleles arose early during treatment but did not sweep the gut population, indicating that other genetic mutations reducing caspofungin susceptibility had also arisen in the same populations, improving their relative fitness to make them competitive with *fen1-W39** or *fks2-R1378H*. We also provide evidence that in both clinical and mouse gut-evolved *C. glabrata* strains, *fks* and *fen1* mutations can be acquired sequentially in the same cells (e.g., in GI-297, WM_18.26, and CAS14-5989). Although clinical histories are not always available, the identification of clinical strains containing both mutations in *FEN1* and in *FKS1* or *FKS2* strongly suggests that those mutations had arisen under the same environmental pressure. Together, our results are consistent with a model for evolution of caspofungin resistance in the gut where caspofungin treatment rapidly kills the majority of cells in the gut (which can be seen as the several-log drop in fungal burdens, Fig 2A), but rare drug-tolerant cells survive (Fig 7). These cells may be caspofungin-tolerant either due to pre-existing mutations or to non-genetic mechanisms [56]. Such tolerant cells preferentially proliferate in the gut, and additional mutations further improving fitness in the presence of caspofungin may arise during this proliferation, perhaps stimulated by stress-induced mutagenesis [57,58]. Weak, incremental improvements in fitness can be achieved by a number of genetic alterations (e.g., any loss-of-function mutation in *FEN1* or a weak mutation in *FKS2*) and can therefore occur at a relatively high frequency. The occurrence of these early mutations allows sufficient numbers of *C. glabrata* cells to survive and proliferate in the gut during caspofungin treatment to eventually develop a rare, clinically echinocandin-resistant mutation in *FKS2* (e.g., S663P), which may then take over the population and eventually cause an echinocandin-resistant systemic infection.

There are several potential reasons why the gut may provide a permissive environment for the evolution of echinocandin resistance. First, drug concentrations reached in the GI tract during treatment are several times lower than those reached in the plasma and various organs

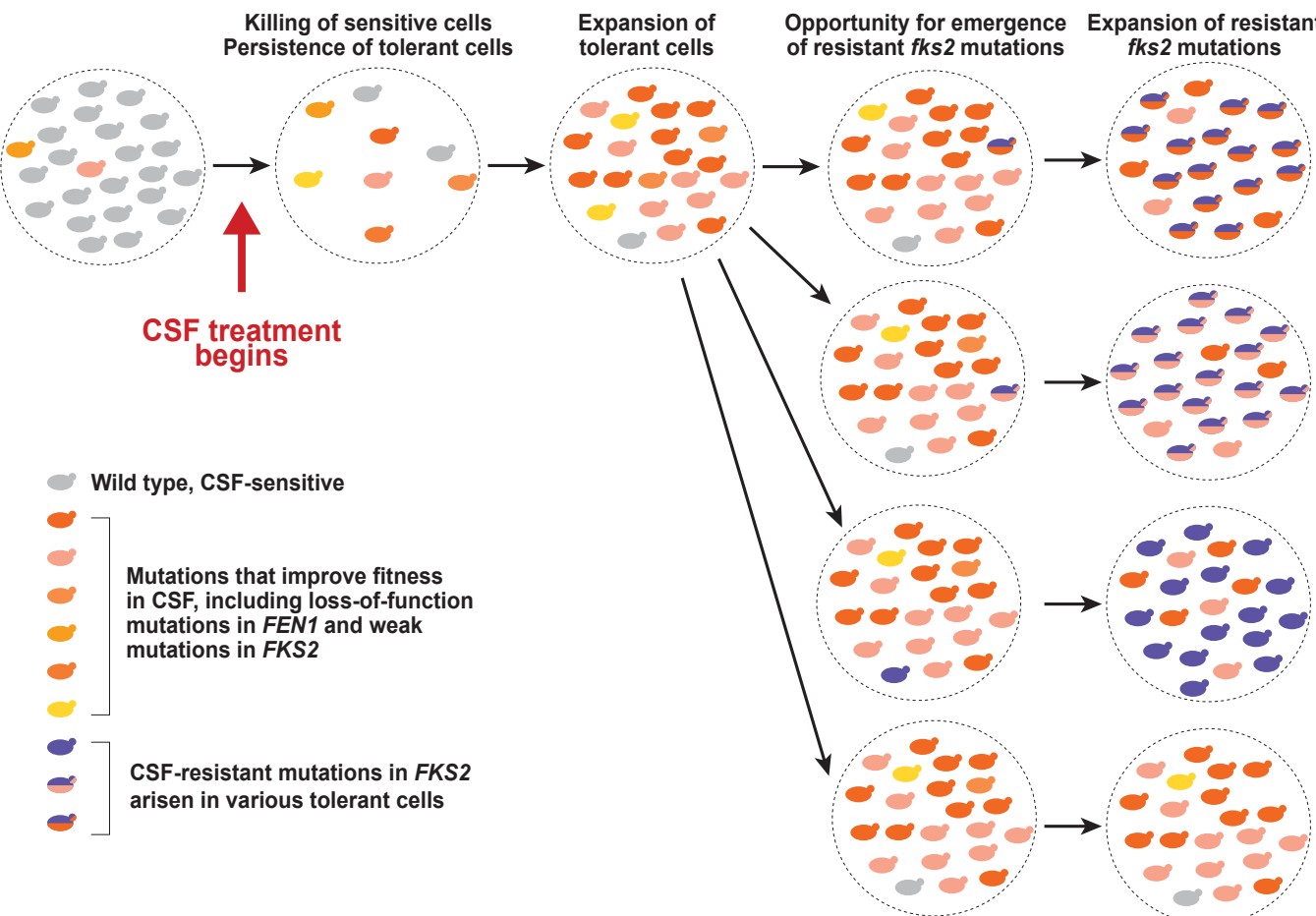

**Fig 7. Model for evolution of caspofungin (CSF)-resistant *C. glabrata* in the gut reservoir during CSF treatment.** Our results support the model where, upon the initiation of CSF treatment, CSF-sensitive cells are killed but rare CSF-tolerant cells expand in the gut. This CSF-tolerant cells may be due to pre-existing rare mutations or to non-genetic mechanisms. Such CSF-tolerant cells preferentially expand in the gut, and additional mutations with increased fitness during CSF treatment are further selected and expanded. At this stage the gut is a genetically heterogeneous population of cells with increased fitness in the presence of CSF, such as those with mutations in *FEN1* or weak mutation in *FKS2*. Ultimately, this survival and proliferation of the gut *C. glabrata* reservoir during CSF treatment may culminate in the formation and preferential expansion of a clinically CSF-resistant mutation, such as *fks2*^S663P, which has the highest fitness in the presence of the drug and may eventually cause an echinocandin-resistant systemic infection.

[37]. Second, fungi form prominent biofilms in the GI tract, which are more tolerant to anti-fungal treatments [59]. Either of these possibilities is consistent with our observation that *fen1* mutants have a fitness advantage over wild-type cells in the gut but not in the kidney, where caspofungin concentrations are high enough to eliminate both *fen1* and wild-type strains equally well, whereas a *fks2* hot-spot mutation with a stronger resistance phenotype is killed less efficiently. Thus, a paradigm emerges wherein in the context of disseminated infection, echinocandin treatment effectively eliminates infecting circulating strains with moderately reduced drug susceptibility (such as *fen1*), whereas in the context of GI colonization, the same strains survive during treatment, maintaining a continuous viable *C. glabrata* reservoir in which stronger *fks* mutations can then emerge (Fig 7), leading to therapy failure. This scenario may be different for the new β-glucan synthase inhibitor ibrexafungerp, which is orally bio-available and therefore may be present in the GI tract at concentrations high enough to elimi-nate the gut-colonizing *C. glabrata* population, but this remains to be experimentally verified.

Several previous studies have reported that mutations in the sphingolipid pathway genes, including *FEN1*, reduce fungal susceptibility to caspofungin but not to other echinocandins *in*

*vitro* and that these mutations cause an increase in cellular PHS levels [41,43,49,50]. Our study builds on those results and reveals several key new mechanistic insights. First, we demonstrate that genetic depletion of PHS by deletion of alkaline ceramidase-encoding gene *YPC1* sensitizes *C. glabrata* to caspofungin, strengthening the conclusion that phytosphingosine levels are a key determinant of caspofungin susceptibility. Furthermore, *ypc1Δ* was fully epistatic to *fen1Δ*, confirming that increased PHS is the cause of reduced caspofungin sensitivity in the *fen1Δ* strain. Second, we show that caspofungin association with the cellular surface is significantly reduced in the *fen1Δ* mutant but is restored in the *fen1Δ ypc1Δ* mutant, correlating with their reduced and restored caspofungin sensitivity, respectively. This result, as well as our MS analysis of membrane-associated CSF, suggests that the reduced caspofungin efficacy in the *fen1Δ* mutant is underpinned by its reduced capacity to bind to the plasma membrane due to the membrane's altered lipid composition (i.e., higher phytosphingosine content), which in turn likely decreases caspofungin ability to inhibit membrane-embedded glucan synthase. A similar hypothesis was proposed by Healey et al. [43], suggesting that the accumulation of long chain bases, such as phytosphingosine, in the plasma membrane reduced the association of caspofungin, but not other echinocandins, with glucan synthase. Further studies are necessary to fully elucidate the effect of sphingolipid pathway mutations on glucan synthase and its interaction with different echinocandin class drugs.

Our results provide one potential explanation for frequent observations of clinical strains showing differential susceptibility to different echinocandin class drugs [15–18,29]. Specifically, based on our results and previous *in vitro* studies [43], it is likely that strains showing non-susceptibility only to caspofungin are likely to carry mutations in the sphingolipid biosynthesis pathway or, perhaps, mutations in other pathways that cause the accumulation of PHS. We further speculate that other, as yet unknown, pathways differentially affect *C. glabrata* susceptibility to micafungin and anidulafungin, and that studying the evolution of resistance to these drugs in the mouse gut colonization model can help identify clinically-relevant mutations that reduce *C. glabrata* susceptibility to one echinocandin while not affecting, or increasing, its susceptibility to the other echinocandins. If such mutants were identified, this information could have important therapeutic implications. For instance, the fact that the *fen1* mutations arising in gut-colonizing *C. glabrata* are highly susceptible to micafungin suggests that, contrary to the conventional wisdom of not combining drugs belonging to the same class, using a combination of caspofungin and micafungin may be of value because it would constrict the pathogen's evolutionary trajectory towards pan-echinocandin-resistant *fks* mutations.

Our conclusions about diverse *C. glabrata* strains coexisting in the gut are highly consistent with genomic analyses of sequential clinical isolates from human patients, which show that a single individual can carry multiple closely related strains, including multiple drug-resistant variants that emerge during drug therapy [24,60]. Together, these observations in the mouse model and in human candidiasis patients underscore the capacity of *C. glabrata* to undergo micro-evolution in the host and may explain the very high genetic diversity among clinical strains [30, 35]. This capacity for evolution makes *C. glabrata* a formidable clinical challenge; nevertheless, understanding the mechanisms promoting its evolvability within relevant host niches may help develop effective measures to reduce the evolution of drug-resistant strains.

## Materials and methods

### Ethics statement

All animal study protocols used in this work have been approved by the Hackensack Meridian Health Center for Discovery and Innovation Institutional Animal Care and Use Committee under protocol number 262.

## Yeast strains and media

The *C. glabrata* strain used in the mouse conization experiments was DPL1021 (ATCC90030) and the described deletion mutants constructed in the DPL1021 background. Cells were cultured in standard yeast extract-peptone-dextrose (YPD) medium at 37°C, which is the optimum growth temperature for this species. Deletion mutants were generated in-house using a CRISPR-CAS9 targeted integration replacing the desired ORF by a nourseothricin (NAT)- or hygromycin (HYG)-resistance cassette. The deletion construct containing the NAT-resistance or HYG-resistance cassette flanked by regions homologous to the locus of interest was amplified from genomic DNA using primers listed in S5 Table. Integration or the deletion cassettes was performed using CRISPR as described previously [61]. Transformants were selected on nourseothricin (NAT)- or hygromycin (HYG)-containing plates and validated by PCR amplification and sequencing of the targeted locus using external primers (S5 Table). At least two independent transformants were generated and analyzed for every deletion mutant. Primers were ordered from Integrated DNA Technologies (Coralville, IA, United States) and Azenta (South Plainfield, NJ, United States), and all Sanger sequencing of the above-described constructs was done by Azenta (South Plainfield, NJ, United States).

## Murine model of *Candida glabrata* GI colonization

The GI model of *C. glabrata* colonization was performed as described in [40] with some modifications. 6-week-old female CF-1 immunocompetent mice (Charles River Laboratories) treated subcutaneously, daily from day-2 to 15, with 320 mg/kg of piperacillin-tazobactam (PTZ, 8:1 ratio, AuroMedics Pharma LLC, East Windsor, NJ, United States) to clear native intestinal bacterial microbiota. On day 0, mice were inoculated via oral gavage with approximately $1.5 \times 10^8$ CFU of *C. glabrata* in 0.1 ml of PBS. In this manner mice were colonized with strain DPL1021 (a.k.a. ATCC90030) or its derivative mutants. Daily intraperitoneal administration of 20 mg/kg of caspofungin (Selleck chemicals) or PBS was initiated on day 3 post inoculation and continued through day 15. Fresh fecal samples were collected every other day throughout the experiment to assess fungal burden in the GI tract. One mouse was included as an untreated control (treated with PBS alone). All mice in the PBS alone control groups maintained a constant fungal burden of $10^6$–$10^8$ CFU/g of stool. The mice were housed individually to avoid horizontal transfer of *C. glabrata* strains.

## Competitive fitness in the GI tract

For the mixed colonization fitness study, mice were gavaged with an equal mix of *C. glabrata* DPL1021 (ATCC90030) wild type and isogenic *fen1Δ* or *fen1*-W145* mutants totaling $1.5 \times 10^8$ CFU in 0.1 ml of PBS. In each group, four mice were treated daily with caspofungin as described above and four other mice were used as an untreated control (treated with PBS alone). Fecal samples were plated on the indicated days and the *FEN1* status of the colonies determined as follows. To distinguish *fen1Δ* from WT colonies, the plates were replica-plated to YPD plates containing 250 μg/ml nourseothricin. To distinguish *fen1*-W145* from WT colonies, colony PCR using a molecular beacon was performed as described below.

## Asymmetric PCR and molecular beacon-based melting curve analysis

One set of primers was designed to amplify the W145* region of *fen1* on *C. glabrata*. The *fen1* region was amplified by using excess primer *fen1*-W145-RV (5'-GAGTTTATTGA-CACCCTCTTCTTGG-3') and limiting primer *fen1*-W145-FW (5'-GATCTACCAT-CACGGTTTATTCTATGCC-3'). The molecular beacon (MB) was designed by adding two

artificial arm sequences to both ends of the target sequence. The MB was labeled with the fluorophores 5-carboxyfluorescein (FAM) at 5' and tetramethylrhodamine (TAMRA) at 3' end, targeting the W145* allele sequence of *fen1*, and it's: (5'-FAM-TTG-CAAATTGGCTTGCCCCCACT-TAMSp-3'). The secondary structure of the MB was evaluated by the software OligoAlanyzer 3.0 (http://www.idtdna.com/analyzer/Applications/OligoAnalyzer/). Asymmetric PCR was carried out on the AriaMx real-time PCR system (Agilent Technology, CA) in 20 μl reaction volume using SensiFAST probe no-ROX mix (Bioline, London, UK). The *fen1* duplex assay contained 5 μM *fen1-145*-FW, 20 μM *fen1-145*-RV and 5 μM the MB. Instead of using genomic DNA, we used a single colony of *C. glabrata* to discriminate between the DPL1021 (ATCC90030) wild type strain and the *fen1-W145Stop* mutant. The PCR conditions were 95˚C for 3 min; 45 cycles of 10 s at 95˚C, 20 s at 60˚C, and 30 s at 72˚C; and 2 min at 72˚C. Immediately after amplification, melting curve analysis was initiated as a minute incubation at 95˚C, after which it was melted from 53˚C to 64˚C with a ramp rate of 0.1˚C/s.

### Systemic infection

The systemic infection study was performed essentially as described [56] with some modifications using outbred CD-1 mice weighing 22–24 g (Charles River Laboratories). Neutropenia was induced and maintained through administration of 150, 100, and 100 mg/kg cyclophosphamide via IP injection on days -4, -1, and 2 post infection, respectively. Groups of mice were randomized into 3 inoculation arms infected with WT (DPL1021/ATCC90030) and *fks2-S663P* and *fen1Δ* mutants in the same strain background. On Day 0, mice were inoculated with 0.1 ml of a cell suspension containing $1x10^7$ CFU/mouse via Retro-Orbital 50 μl. Within each inoculation arm, mice were randomized into the antifungal treatment sub-groups (caspofungin 1.25, 2.5, and 5 mg/kg i.p.) and into the vehicle treatment (PBS i.p.) sub-group. The numbers of mice used are shown in Fig 2C (each dot indicates one mouse). Antifungal therapy was started at 24h post-infection and continued through day 2 (2 days total), during which caspofungin and vehicle control were administered once daily. On Day 3, the mice were euthanized via $CO_2$ narcosis and kidneys were aseptically harvested for CFU count. Both kidneys were placed in M tubes containing 2.5 ml PBS, homogenized and plated onto YPD plates for kidney burden counts.

### Echinocandin susceptibility testing

Echinocandin susceptibility testing was performed using a broth microdilution method following CLSI standards (Clinical Laboratory Standards Institute (CLSI), 2017) with some modifications. The media used was 2X RPMI and the final concentrations tested ranged from 0.015 to 2 μg/ml in two-fold increasing concentrations. Minimum inhibitory concentrations (MICs) were visually read after 48 h of incubation at 37˚C and at least three biological replicates were performed. CSF and CSF-AF647 susceptibilities (S1B Fig) were measured side by side using the concentrations shown in the figure. In that experiment all wells contained 2% DMSO to match the well with the highest CSF-AF647 concentration (205 μg/ml) and growth was assessed after 48h by measuring optical density at 800 nm because CSF-AF647 does not absorb at this wavelength.

### Lipidomic analysis

Yeast were precultured, diluted to 0.3 $OD_{600}$ unit/ml and cultured to 1 OD unit/ml, then treated with caspofungin (0.5 μg/ml) for 1 hour. The cells were washed and stored at -20˚C. The lipidomic analysis was done at the lipidomic resource of the Medical University of South Carolina. Inorganic phosphate (Pi) levels were measured in the same samples and all lipids were normalized to Pi.

## Multi-locus Sequence Typing of *C. glabrata*

STs were assigned by using the MLST scheme based on 6 loci (*FKS2*, *LEU2*, *NMT1*, *TRP1*, *UGP1*, *URA3*) according to Dodgson et *al.*, 2003 [54].

## Spot assays

*C. glabrata* cells grown overnight were washed twice with phosphate-buffer saline (PBS) and adjusted to $5x10^5$ CFU/ml in PBS. An equal volume (3 μl) of 10-fold serial dilutions of each strain were spotted onto YPD plates either lacking drug or containing the indicated concentrations of caspofungin. The plates were incubated at 37˚C for 2 days.

## Super-resolution microscopy

Cells were cultured overnight in YPD broth, then washed twice and resuspended in Synthetic Defined (SD) Medium to $OD_{600}$ = 0.3 and allowed to grow to approximately $OD_{600}$ = 1. Subsequently, the cells were incubated with 0.26 μM CSF-AF647 (WuXi Biologics) for 30 minutes in the dark at 37˚C. The cells were then washed twice with PBS, fixed with 2% paraformaldehyde, and washed twice more with PBS. For imaging, the cells were dispersed on u-Slide 8 well high glass bottom (iBidi, Cat.No: 80807) that had been pre-treated with Poly-L-Lysine (ScienceCell, Cat.No: 0403). The imaging chamber was overlaid with 100 ul of BCubed A and 1ul of BCubed B dSTORM buffers (Oxford Nanoimager, ONi). Visualization of the cells was performed using the Nanoimager (Oxford Nanoimaging, ONi) using the 640 nm laser configuration for CSF-AF647, utilizing a 100x oil-objective lens. The images were analyzed using CODI software.

## Whole genome sequencing of gut-evolved *C. glabrata* strains

Genomic DNA of gut-evolved *C. glabrata* strains was extracted using the Quick-DNA Miniprep Kit (Zymo Research). Library preparation and next-generation sequencing was conducted by Azenta (Plainfield, NJ). Preliminary sample quality check was performed using FastQC version 0.12.0 [62], only runs with a median Q > 30 where further analyzed. Reads were aligned to DPL1021 (ATCC 90030) reference genome using STAR version 2.7.1 [63], aligned reads where sorted and then indexed using samtools version 1.18 [2], alignment metrics where also generated using samtools using flagstat command, variant calling was done using bcftools version 1.18 [64] combining mpileup and call commands under standard parameters, variants where deduplicated using Picard version 3.1 f (Picard Toolkit, Broad Institute, 2019) using MarkDuplicates command and the alignment metrics previously generated, reads where further annotated using snpsift version 5.1 [65] and filtered by effect severity using snpeff version 5.1 [66]. The genome sequencing data have been deposited as NCBI PRJNA1010358.

## Identification of unique SNPs in *FEN1*, *FKS1*, and *FKS2* in publicly available genomes

488 paired end read files from 244 runs were downloaded from NCBI PRJNA524686 [67] and 118 paired end read files from 59 runs were downloaded from PRJNA480138. Multi Locus Sequence Typing (MLST) was done by matching sequences of 'FKS2', 'LEU2', 'NMT1', 'TRP1', 'UGP1', 'URA3' loci to reference alleles [54] (https://pubmlst.org/organisms/candida-glabrata). SNPs in *FEN1*, *FKS1*, and *FKS2* were identified using the variant calling method described above.

## Amplicon sequencing

Fecal samples were plated on YPD plates containing piperacillin and tazobactam to reduce bacterial contamination. Approximately 1000 colonies were collected and pooled together for each sample. In cases where only a limited number of colonies grew from the first plating (e.g., mouse 3 day 9, see S2A Fig) the remainder of the fecal sample was plated to obtain the desired number of colonies. Samples from mouse 2 day 7 and mouse 4 day 15 were not obtained because of extensive mold and bacterial contamination on the YPD plates. Genomic DNA was extracted from the pooled colonies using phenol-chloroform extraction. For the sequencing of *FKS1* and *FKS2* hotspot regions (HS1 and HS2), primer pairs were designed to encompass each of the four hotspots, generating amplicons of 200 to 300 bp. To sequence the entire *FEN1* ORF, 6 overlapping amplicons, ~300bp each, were also generated by PCR using the Q5 High-Fidelity MasterMix (New England Biolabs), followed by purification with a DNA sequencing clean-up kit (Zymo Research). All primers are listed in S5 Table. DNA concentrations was quantified using the Qubit system (ThermoFisher). The 10 amplicons from each sample were mixed together in approximately equimolar concentrations and amplicon sequencing was performed at CD Genomics (Shirley, NY 11967, USA). SNPs were identified using the variant calling method described above. The percentage of cells containing a mutation was approximated as the number of reads containing a SNP divided by the number of reads aligning to that region on the deduplicated, aligned and sorted read file for each run. The amplicon sequence reads have been deposited as NCBI PRJNA1010358.

## Analytical method for CSF quantification in CSF-AF647 by LC-MS/MS

DMSO stocks of 10.27 mg/ml CSF-AF647 conjugate were diluted in extraction solution (1:1, MeCN:MeOH) containing verapamil (internal standard). 1 mg/ml DMSO stock of CSF (Selleck chemicals) was used to create calibration standards for quantification of CSF in the CSF-AF647 conjugate stock. Working solutions covering the desired concentration range were prepared by diluting the stock solutions in extraction solution. Samples were vortexed for 5 minutes and 150 µl was transferred to a 96 well plate for analysis LC-MS/MS analysis was performed on a SCIEX QTRAP 6500+ triple-quadrupole mass spectrometer coupled to a Shimadzu Nexera UHPLC system. Chromatography was performed on an Agilent SB-C8 (2.1 x 30mm; particle size 3.5 µm) using a reverse phase gradient with a flow rate of 0.6 mL/min. MQW deionized water with 0.1% formic acid (FA) was used for the aqueous mobile phase and 0.1% FA in MeCN for the organic mobile phase. Multiple-reaction monitoring (MRM) of precursor/fragment transitions in electrospray positive-ionization mode was used to trace CSF and verapamil. MRM transitions of 1093.70/1033.50 and 455.40/165.00 were used for CSF and verapamil respectively. Data processing was performed using the Analyst software (version 1.7.2 Applied Biosystems Sciex).

## Crude membrane preparation and CSF quantification by LC-MS/MS

Exponentially growing *C. glabrata* cells were incubated with 0.25 µg/ml CSF for 20 min at 37°C, then harvested by centrifugation, washed twice with 1 ml cold PBS, and resuspended in 0.5ml of disruption buffer (50 mM Tris-HCl, pH 7.6, 150mM sodium chloride, 1 mM dithiothreitol, 20% glycerol, and protease inhibitors (Roche Complete Ultra)). The cells were homogenized in Lysing matrix E tubes (MP Bio 116914100) by BeadBlaster 24 Microtube Homogenizer (Benchmark Scientific). Beads and debris were removed by centrifugation at 3500g for 5 min, then the supernatants were centrifuged at 17,000g for 15 min at 4°C to obtain membrane pellets. The membranes were resuspended in 100 µl disruption buffer, 10 µl was saved for protein measurement via Bradford assay, and 90 µl was pelleted again and snap

frozen. Prior to LC-MS/MS analysis, 200 μl of extraction solution (1:1, MeCN:MeOH) containing verapamil (internal standard) was added to membrane pellets. Extracts were sonicated for 5 minutes, vortexed for 5 minutes with beads and centrifuged at 4,000 rpm for 5 minutes. 150 μl of supernatant was transferred to a 96 well plate for analysis. The LC-MS/MS analysis was performed as described above.

### Statistical analyses

Statistical analyses were performed using GraphPad Prism software, applying the appropriate statistical tests described in the corresponding figure legends.

### Disclaimer

The findings and conclusions of this report are those of the authors and do not necessarily represent the official position of the Centers for Disease Control (CDC).

### Supporting information

**S1 Fig. Structure and synthesis scheme of Alexa Fluor 647-conjugated caspofungin (CSF).** **A.** CSF-AF647 association with the fungal cells was significantly reduced in *fen1Δ* cells in ATCC90030 strain background. **B.** Alexa Fluor 647-conjugated CSF (CSF-AF647) showed reduced antifungal activity, which was further significantly reduced in strains carrying the echinocandin-resistant mutation *fks2-S663P* and *fen1Δ*. Growth was measured as $OD_{800}$ to avoid interference from absorbance by CSF-AF647. **C.** AF647 alone does not bind to *C. glabrata* cells, unlike CSF-AF647. The amine-reactive NHS ester of AF647 (Lumiprobe) was quenched in 10X excess TRIS and the molecule was incubated with *C. glabrata* cells at the same molar concentration and for the same amount of time as CSF-AF647, followed by ONi imaging (see Methods).
(EPS)

**S2 Fig. *C. glabrata* gut colonization for amplicon sequencing. A.** Fungal burdens per gram of stool in the GI tracts of four mice used in the amplicon sequencing experiments. **B.** Mutations in *FKS1* hot-spots 1 and 2 present at low levels in the original culture used to gavage the mice in the amplicon sequencing experiment. The same mutations were recovered at approximately the same levels from the colonized mice during CSF treatment. Samples from mouse 2, day 7 and mouse 4, day 15 were not obtained due to heavy bacterial and mold contamination of the plates.
(EPS)

**S1 Table. Table of coding SNPs identified in the gut-evolved *C. glabrata* strains.**
(XLSX)

**S2 Table. Levels of lipids analyzed as part of the ceramide-sphingolipid panel in wild type and *fen1* mutants.** These results were used to make the heatmap in Fig 4A. The levels were normalized to inorganic phosphate measured in the same samples (arbitrary units).
(XLSX)

**S3 Table. *FKS1*, *FKS2*, and *FEN1* coding SNPs found in the genomes of clinical *C. glabrata* isolates sequenced and made publicly available by the CDC (https://www.ncbi.nlm.nih. gov/biosample?Db=biosample&DbFrom=bioproject&Cmd=Link&LinkName=bioproject_ biosample&LinkReadableName=BioSample&ordinalpos=1&IdsFromResult=524686).**
(XLSX)

**S4 Table.** *FKS1*, *FKS2*, and *FEN1* coding SNPs found in the clinical C. glabrata genomes from Biswas et al., 2018 [30].
(XLSX)

**S5 Table. Primers used in this study.**
(XLSX)

## Acknowledgments

We thank research and administrative staff at the Centers for Disease Control for providing information on clinical *C. glabrata* strains [67]. We also thank Candidemia staff from the Georgia Emerging Infections Program, Maryland Emerging Infections Program, and Tennessee Emerging Infections Program; and participating Emerging Infections Program Laboratories. Finally, we thank Dr. Milena Kordalewska for help with designing the *fen1-W145\** molecular beacon, Tara Lozy for assistance with statistical analysis, and the CDI shared flow cytometry and microscopy cores for assistance with the corresponding experiments.

## Author Contributions

**Conceptualization:** Erika Shor.

**Data curation:** Ariel A. Aptekmann, Diego H. Caceres, Nancy A. Chow.

**Formal analysis:** Ariel A. Aptekmann.

**Funding acquisition:** David S. Perlin, Erika Shor.

**Investigation:** Yasmine Hassoun, Mikhail V. Keniya, Rosa Y. Gomez, Nicole Alayo, Giovanna Novi, Christopher Quinteros, Firat Kaya, Matthew Zimmerman.

**Methodology:** Yasmine Hassoun, Ariel A. Aptekmann, Mikhail V. Keniya, Firat Kaya, Matthew Zimmerman.

**Supervision:** Matthew Zimmerman, David S. Perlin, Erika Shor.

**Validation:** Mikhail V. Keniya.

**Writing – original draft:** Yasmine Hassoun, Erika Shor.

**Writing – review & editing:** David S. Perlin, Erika Shor.

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
