## [Decision Letter · Decision Letter 0]

21 Dec 2023

Dear Dr. Shor,

Thank you very much for submitting your manuscript "Evolutionary dynamics in gut-colonizing Candida glabrata during caspofungin therapy: emergence of clinically important mutations in sphingolipid biosynthesis" (PPATHOGENS-D-23-02041) for consideration at PLOS Pathogens. As with all papers peer reviewed by the journal, your manuscript was reviewed by members of the editorial board and by several independent peer reviewers. Based on the reports, we regret to inform you that we will not be pursuing this manuscript for publication at PLOS Pathogens.

The reviewers thought that the topic was interesting and noted the importance of studies on the evolution of drug resistance in Candida glabrata. However, they also had significant concerns that limited their enthusiasm for the manuscript. In particular, Reviewers 1 and 2 thought that the previous reports that a fen1 mutation causes Caspofungin resistance in C. glabrata and other fungi undercut aspects of the novelty and significance. Both of these reviewers stated that further research would be needed to advance the understanding of the mechanisms by which altered PHS affects Caspofungin susceptibility in order to increase the significance of these studies. Reviewer 1 also had several concerns about the limitations of the data supporting the interpretations of the dynamics of the evolution of Caspofungin resistance. For one, they thought that there were too few mice and too few colonies studied to make significant conclusions. Furthermore, they thought that the emergence of the fen1 mutant strains at early times but then disappearance at later times raised further questions. Both Reviewers 1 and 2 also had concerns about some experiments involving the use of the evolved strains, which may carry additional mutations. In light of these serious concerns, additional studies would need to be done to make the manuscript acceptable for publication.

The reviews are attached below this email, and we hope you will find them helpful if you decide to revise the manuscript for submission elsewhere. We are sorry that we cannot be more positive on this occasion. We very much appreciate your wish to present your work in one of PLOS's Open Access publications. 

Thank you for your support, and we hope that you will consider PLOS Pathogens for other submissions in the future.

Sincerely,

Michal Olszewski

Section Editor

PLOS Pathogens

Kasturi Haldar

Editor-in-Chief

PLOS Pathogens

orcid.org/0000-0001-5065-158X

Michael Malim

Editor-in-Chief

PLOS Pathogens

orcid.org/0000-0002-7699-2064

Reviewer's Responses to Questions

**Part I - Summary**

Reviewer #1: In this study, the authors investigated the evolutionary dynamics of C. glabrata colonizing the gut of mice during caspofungin (CSF) treatment. Utilizing whole genome and amplicon sequencing, they observed a rapid emergence of both drug target (FKS2) and non-drug target mutations, resulting in reduced susceptibility to CSF. About half of the mutants carry mutations in the FEN1 gene, which encodes a fatty acid elongase involved in sphingolipid biosynthesis. The authors further demonstrated that the fen1∆ and fen1-W145* mutants completely outcompeted the wild-type strain in the gut during CSF treatment. Lipidomics analysis detected significantly increased intracellular levels of phytosphingosine, a sphingolipid biosynthesis intermediate. Blocking phytosphingosine synthesis by deleting the YPC1 gene, which encodes an alkaline ceramidase, hypersensitized the wild-type strain to caspofungin and was epistatic to fen1Δ. Additionally, several different fen1 mutations were identified in clinical C. glabrata isolates, and all of them were shown to phenocopy the fen1Δ mutant, causing reduced CSH susceptibility.

The data presented do not support the authors’ claim that ‘These studies reveal new genetic and molecular determinants of clinical caspofungin susceptibility and illuminate the dynamic evolution of drug target and non-drug target mutations reducing echinocandin efficacy in patients colonized with C. glabrata.’. First, various aspects of the main findings presented in this manuscript have been reported in other fungi including S. cerevisiae, Aspergilus, C. albicans, and C. glabrata. Second, the experiments investigating the evolution dynamics was not well-designed.

Previous findings include:

1. Disruption of FEN1 reduces cellular sphingolipid levels and results in the accumulation of the long chain base, phytosphingosine. (JBC 272: 17376; 1997).

2. Sharma S, et al. (2014) Sphingolipid biosynthetic pathway genes FEN1 and SUR4 modulate amphotericin B resistance. Antimicrob Agents Chemother 58(4):2409-14 (In budding yeast).

3. Increased resistance to caspofungin was first reported in C. glabrata fen1Δ by Healey et al. more than 10 years ago, which was attributed to increased cellular levels of some sphingolipid biosynthesis intermediates, including phytosphingosine (Mol Microbiol 86:303; 20120). Similar results were also reported for Candida albicans fen1Δ fen12Δ mutant and Aspergillus nidulans (basA mutant) in another paper by Healey et al. (Antimicrob Agents Chemother 59:3377; 2015). While the authors cited these papers, they failed to explicitly describe the previous findings in the Introduction.

4. Furthermore, a recent paper by Gao et al. demonstrated significant increase in the cellular levels of sphingolipid biosynthesis intermediates in Candida albicans fen1Δ fen12Δ mutants (Chew et al. Nat Commun 9:4495; 2019), significantly reducing susceptibility to fluconazole, and deleting the transcription factor that promote key genes in the synthesis pathways abolished the resistant phenotype.

Reviewer #2: This manuscript examines the appearance of echinocandin resistance in Candida glabrata (Cg). The primary finding is of a strong link between biosynthesis of sphingolipids and resistance to echinocandin antifungal drugs. Loss of function mutations in a gene called FEN1 leads to the accumulation of a precursor to ceramide production (PHS) with an accompanying decrease in echinocandin susceptibility. The authors also demonstrate that loss of the ceramidase enzyme Ypc1 (breaks down ceramide into PHS and very long chain fatty acid constituents) both increased susceptibility to caspofungin and restored susceptibility to a fen1 null strain. A gut colonization model was used to demonstrate that fen1 mutants could be selected during in vivo growth of Cg during caspofungin treatment. Additionally, sequencing of clinical mutants revealed that fen1 mutations could be found in these isolates. The authors also used plasmid complementation to analyze the functionality of several of these clinical mutant alleles of FEN1 and provided an argument that these all represented loss of function forms of Fen1.

This is nice work and succinctly presented. The dynamic analysis of caspofungin treatment on Cg resistance in a mouse exposed to caspofungin using amplicon sequencing is well done.

Reviewer #3: This is a well-written manuscript that describes mutations in FEN1 in Candida glabrata that impart increased fitness in the setting of gut colonization and caspofungin treatment. Echinocandin resistance has emerged in candida species, including C. glabrata and in this species is typically attributed to mutations in the gene encoding the drug target glucan synthase (FKS2). However, resistance that cannot be explained by such mutations has been observed. Moreover, it is well established that the gut serves a reservoir for the emergence of echinocandin resistance. This group previously evolved echinocandin resistant strains of C. glabrata using the mouse gut colonization model during caspofungin treatment and observed FKS2 mutations in strains exhibiting increased MICs to both caspofungin and micafungin. However, there were several strains that exhibited increased MICs to only caspofungin and that had no FKS2 mutations. These are the focus of the current study where after WGS, mutations in other genes were observed, most commonly mutations in FEN1 encoding fatty acid elongase such as that leading to the W145* amino acid substitution that should result in a loss of function. Introduction of this mutation or deletion of FEN1 resulted in strains with enhanced fitness in the mouse gut under caspofungin treatment. As opposed to the gut where caspofungin concentrations are lower, this was not observed for kidney fungal burden in a systemic model of infection where caspofungin concentrations are higher. Elevated phytosphingosine was observed in the fen1 KO mutant as well as one gut evolved strain with a FEN1 mutation and one without, suggesting this accumulation may drive this phenotype. This was supported by hypersusceptibility observed when YPC1 (alkaline ceramidase) was deleted. Experimental evidence further indicated that the FEN1 mutations arise before FKS2 mutations suggesting the increased fitness conferred by FEN1 mutations may provide a jumping off point for the evolution of fulminant resistance conferred by FKS2 mutations. This was supported by the observation of FEN1 mutations in clinical isolates with reduced susceptibility to caspofungin. In short, these data support a role for FEN1 as a genetic determinant of reduced susceptibility to caspofungin in C. glabrata.

This paper is significant in my opinion for two reasons. First, it gets at the question of factors that contribute to the emergence of FKS2 mutations and caspofungin resistance specifically as only such mouse experiments could. Secondly, it underscores the need to understand more than just classical genetic determinants of antifungal resistance, but also those genetic determinants of more subtle changes in susceptibility that might represent stepping stones for the emergence of fulminant resistance. Such information might ultimately lead to strategies to mitigate the emergence of resistance. 

The introduction is clear and concise with the needed background, the results are clearly presented with good use of figures and tables, and the discussion is a sound treatment of the findings. Methods are clearly presented and appropriate in my opinion.

**Part II – Major Issues: Key Experiments Required for Acceptance**

Reviewer #1: 1. The manuscript extensively discusses the evolution dynamics of CSF-resistant mutants based on the results shown in Figure 4. However, the small sample sizes, i.e., two colonies per day and a total of six mice, are not sufficient to produce any credible evolution patterns. The detection of various mutations in the population on different days appears random, raising more questions than answers. For example, while panel A shows that all fen1 mutations appeared before fks2 mutations, panel B shows the opposite. There is no evidence that the earlier appearance of tolerance or resistance due to fen1 mutations has allowed time for FKS2 resistance to evolve. Although mutations were detected in both fen1 and fks2 in the two colonies from mouse #1 on day 15, the fen1 mutation differs from the one detected on day 9. The fen1-D166N mutant in mouse #2 must be extremely dominant, as it was detected in both colonies from day 7 to day 15; however, no mutations were detected in FKS2. In mouse #4, the two fen1 mutations detected on day 7 have nothing to do with the FKS2 mutations detected from day 9 to day 15. The fen1-Q10stop mutant found on day 7 in the two colonies from mouse #5 is intriguing. The evolved stop codon at the tenth position of the amino acid sequence makes it almost like the fen∆ mutant, which was shown to completely outcompete the WT strain under CSF treatment (Figure 2B). Yet, the fen1-Q10stop mutation was not detected again from day 7 to day 15. While one can argue that unknown mutations can influence the competitiveness of different mutants, significantly larger sample sizes are required to reveal credible evolutionary patterns of various drug-resistant mutations.

2. In Figure 2C, it is not clear why only the strain FEN1-W145stop (correct nomenclature should be fen1-W145stop) was used in this experiment but not fen1∆. These two mutants are not equal because fen1-W145stop carries several other mutations which may have unknown influence on this strain’s fitness. The result shows that the fen1-W145stop mutant exhibits similar sensitivity to CSF as the WT strain during systemic infection in mice. However, only one CSF concentration, 5 mg/kg, was used in this experiment. This is a very high concentration compared with the dose used in humans (50-70 mg per day for adults; assuming the average weight of a person is 60 kg). It is likely that this concentration is significantly higher than the MIC of both strains, masking the difference in susceptibility to CSF between them. The experiment should be repeated using lower CSF concentrations, such as 1 mg / kg. Also, is there an estimate of the CSF concentration in the gut where CSF is IP-administered at 20 mg / kg?

3. The authors used different gut-evolved fen mutants in different experiments. For example, they used GI-240 to generate the data in Figure 2, but GI-299 and GI-240 for Figure 3 and Figure 5, respectively. This raises the question whether different strains exhibit significantly different phenotypes, particularly fitness in the gut under CSF treatment. This is worrisome because many of the evolved strains carry multiple mutations. 

4. Lines 188-189. It is described that ‘one non-fen1 gut-evolved strain (GI-63)’ was included in the lipidomics analysis. However, the strain was not seen in Figure 3A.

Reviewer #2: The problem is that it has already been shown in two other fungi that changes leading to defects in sphingolipid biosynthesis and even specifically loss of FEN1 function leads to decreased echinocandin susceptibility. The work described here does not take things too far beyond this fairly well-established observation. If insight could be provided into the basis of the effect of sphingolipid deficiency on echinocandin resistance, then the impact of this work would be enhanced. For example, is the endocytosis of the Fks proteins influenced by defects in normal sphingolipid production? Is the activity of the Fks proteins enhanced or the ability of echinocandins to inhibit these enzymes reduced?

Reviewer #3: (No Response)

**Part III – Minor Issues: Editorial and Data Presentation Modifications**

Reviewer #1: 1. Figure 1. Column 2: What does ‘genotype’ refer to? I thought all the GI strains are evolved strains and most of them carry mutations. Thus, what does WT mean? Also, the four gene deletion mutants were not explained, and they are not mentioned anywhere else in the text.

2. Figure 2A. I figured that sterilization and no sterilization mean if mice were treated with the antibiotic or not. If so, it’s better to use + or – antibiotic treatment. The use of a single antibiotic cannot sterilize the gut of mice. Antibiotic cocktails are often used but still cannot ensure sterilization.

3. Figure 2A. The near zero CFU on day 7 in three mice inoculated with WT C. glabrata are strange. Excluding these outliers, gut colonization of the WT strain would not be significantly different from that of the mutants. 

4. Line 160. Nomenclature: is W145* the same as W145stop? If so, please use W145* and explain it when it first appears.

5. Line 399, italicize C. glabrata.

6. Line 400, citation format for Healey et al. is different from the rest.

Reviewer #2: (No Response)

Reviewer #3: I have only minor points:

Aside from the original evolved strains, I didn’t see any MICs for other echinocandins. One would expect no change or increased susceptibility to micafungin and anidulafungin in the mutants. Is this observed and does this represent a therapeutically exploitable phenotype? 

Additional discussion of the clinical relevance would be of value in my opinion. While this is interesting and informative biology, there may be ways in which this information might inform how echinocandins might be used to reduce the emergence of resistance.

PLOS authors have the option to publish the peer review history of their article (what does this mean?). If published, this will include your full peer review and any attached files.

Reviewer #1: No

Reviewer #2: No

Reviewer #3: No

---

## [Decision Letter · Decision Letter 1]

7 Jun 2024

Dear Dr. Shor,

Thank you very much for submitting your manuscript "Evolutionary dynamics in gut-colonizing Candida glabrata during caspofungin therapy: emergence of clinically important mutations in sphingolipid biosynthesis" for consideration at PLOS Pathogens. As with all papers reviewed by the journal, your manuscript was reviewed by members of the editorial board and by several independent reviewers. In light of the reviews (below this email), we would like to invite the resubmission of a significantly-revised version that takes into account the reviewers' comments.

The reviewers agree that the manuscript has been significantly improved by the revisions. However, there are still some concerns, especially regarding the new data in Figure 3. 

One concern is that the Halo-tagged Fks1 does not show expected plasma membrane localization. This raises questions about whether the Halo-tagged Fks1 is reporting the true localization of Fks1. In addition to comments from raised by the Reviewer 1, a Western blot should be done to see if there is free Halo that could account for cytoplasmic localization.

There are also questions about the studies using caspofungin covalently conjugated to Alexa Fluor 647 (CSF- AF647). This modified form of caspofungin appears to be ~120-fold reduced in activity, which raises the concern that this level of activity could be due to a small amount of contaminating untagged caspofungin. In addition to addressing comments of the reviewers, it would help if you could determine if there is a significant amount of untagged CSF in the labeled prep. Perhaps this could be done by Mass spectrometry.

Some other controls for the CSF- AF647 need to be performed that are standard for validating this type of labeling experiment. As mentioned by Reviewer 1, one control is to treat cells with AF647 alone to determine if AF647 binds cells independently of caspofungin. Another control is to determine whether the observed binding can be competed with a slight excess of unlabeled caspofungin to determine whether they bind the same target on the cell surface.

We cannot make any decision about publication until we have seen the revised manuscript and your response to the reviewers' comments. Your revised manuscript is also likely to be sent to reviewers for further evaluation.

Please ensure that your revisions are complete to ensure the publication.

Sincerely,

James B. Konopka

Academic Editor

PLOS Pathogens

Michal Olszewski

Section Editor

PLOS Pathogens

Michael Malim

Editor-in-Chief

PLOS Pathogens

orcid.org/0000-0002-7699-2064

The reviewers agree that the manuscript has been significantly improved by the revisions. However, there are still some concerns, especially regarding the new data in Figure 3.

One concern is that the Halo-tagged Fks1 does not show expected plasma membrane localization. This raises questions about whether the Halo-tagged Fks1 is reporting the true localization of Fks1. In addition to comments from raised by the Reviewer 1, a Western blot should be done to see if there is free Halo that could account for cytoplasmic localization.

There are also questions about the studies using caspofungin covalently conjugated to Alexa Fluor 647 (CSF- AF647). This modified form of caspofungin appears to be ~120-fold reduced in activity, which raises the concern that this level of activity could be due to a small amount of contaminating untagged caspofungin. In addition to addressing comments of the reviewers, it would help if you could determine if there is a significant amount of untagged CSF in the labeled prep. Perhaps this could be done by Mass spectrometry.

Some other controls for the CSF- AF647 need to be performed that are standard for validating this type of labeling experiment. As mentioned by Reviewer 1, one control is to treat cells with AF647 alone to determine if AF647 binds cells independently of caspofungin. Another control is to determine whether the observed binding can be competed with a slight excess of unlabeled caspofungin to determine whether they bind the same target on the cell surface.

Reviewer's Responses to Questions

**Part I - Summary**

Reviewer #1: This is a revised manuscript. I thank the authors for their efforts in addressing the issues I raised previously.

The authors added new experiments, where they used fluorescence-labeled CSF to assess CSF’s interaction with the cell surface in the wild type versus the fen1∆ mutant. However, the new data raised some important concerns, which should be addressed.

Reviewer #3: As this manuscript has undergone initial review by three additional reviewers I will keep my assessment brief. This is a well-written manuscript that describes important experiments assessing the in vivo evolution of caspofungin resistance in a murine GI colonization model. These experiments implicate mutations in FEN1 as contributing to the resistant phenotype. Importantly, while much is known about the effect of loss of FEN1 function on echinocandin susceptibility (as pointed out by reviewer 1), this work shows that such loss of function through development of mutations in this gene can be selected for in vivo under caspofungin pressure. This is a key finding. I believe the authors have thoroughly responded to the concerns of reviewer 1 (as well as the other reviewers) and I am supportive of this paper and its importance for the two reasons outlined by reviewer 3. I would also add that the authors then used their experimental observations to explore publicly available WGS data for echinocandin resistant C. glabrata isolates, identified similar mutations in FEN1, and experimentally confirmed that they influence echinocandin susceptibility. This is an important and impactful capstone to this paper in my opinion. I have no additional critique or concerns.

Reviewer #4: This revised manuscript describes experiments that examine the genotypes and phenotypes behind the emergence of caspofungin resistance in gut colonizing Candida glabrata. This is then related to mutations detected in clinical isolates. As earlier reviewer comments state the importance of sphingolipid synthesis pathway and antifungal resistance has been previously elucidated. Here the study goes a step further to unpick the mechanism to explain why sphingolipid levels, in particular phytosphingosine, influence drug susceptibility. The manuscript is substantially strengthened by the inclusion of new tools and the application of high resolution microscopy to analyse the binding of fluorescently-tagged caspofungin to the plasma membrane in gut-evolved and fen1 mutated strains. The microscopy clearly demonstrate that there is less caspofungin binding when PHS levels are increased. The authors include a neat control, the ypc1 mutant, which lowers the PHS levels thus increasing caspofungin binding. This association between caspofungin binding and PHS levels is clear. What is less clear is how these changes in the membrane lipids alter Fks protein abundance at the plasma membrane and/or Fks enzyme activity leading to changes in cell wall glucan levels. The use of the Halo-tagged Fks1 shows clear differences in Fks1 levels and potentially localization between WT and fen1 mutant. In Figure 3D are the zoomed in panels at the bottom overlays of JF549 and AF647 stains? If so would you have expected to see some overlap in their localisation?

The finding that is hard to reconcile is that fen1 mutant doesn’t confer caspofungin resistance in the bloodstream immunocompromised mouse model where kidney fungal burdens were reduced as much as WT cells when mice were treated with range of caspofungin doses (up to 5mg/kg). In the discussion the authors suggest that this may be due to concentrations of bioavailable caspofungin in bloodstream versus the gut. Yet FEN1 mutations do emerge in clinical isolates so there is selection pressure to evolve these mutations in potentially diverse host niches (difficult to ascertain this without detailed information on the clinical isolates). This will require further studies to dissect the impact of host niche on the importance of fen1 mutations.

Minor comments:

Line 123, in summary rather than sum

Line 126 states intracellular levels of PHS, do you mean membrane levels?

Line 148, specify Candida species

Line 155 ergosterol

Line 209 Figure 3D

Figure 2A it is unconventional to use the term sterilization, you explain its meaning in the legend but as you are monitoring fungal counts in faeces you cannot be completely sure that no fungi remain in the GI tract. Why not state caspofungin treatment or plus caspofungin?

**Part II – Major Issues: Key Experiments Required for Acceptance**

Reviewer #1: (1) Figure 3C. Firstly, the results clearly demonstrate that the activity of CSF was reduced by approximately 120-fold (MIC of 0.11 µM for CSF vs. 13 µm for CSF-AF647 against WT) after conjugation with AF647, indicating a significant change in the structure of CSF and loss of activity. Therefore, in contrast to the authors’ statement that ‘it retained significant antifungal activity’ (Line 201), the results show that the conjugate retained very low activity. Secondly, the CSF-AF647 is 98.67% pure. Is the observed low activity of CSF-AF647 due to the presence of a small amount of free CSF molecules or other active intermediates in the sample, i.e., ~1%? Are the 1% impurity and 100-fold decrease of activity a coincidence? Thirdly, there appears to be an issue with the assay of CSF-AF647 activity, as there is no clear concentration dependence in the growth gradient, in contrast to the CSF assay result that shows good gradients. If labeling reduces the activity of a molecule, it is not suitable for studying the molecule’s interaction with a cellular target.

(2) In Figure 3D (CSF-AF647), firstly, the control using AF647 alone is missing. Secondly, a competition experiment should be conducted by adding different concentrations of unlabeled CSF to see whether it reduces the signal of CSF-AF647 on the cell surface in a concentration-dependent manner. Secondly, while Figure 3A shows that HT-FKS1 is functional, which was tested at one SCF concentration, the JF540 staining patterns are very different between the WT and the fen1∆ strain. There are many more bright dots in fen1∆ cells compared to WT cells, and these dots do not seem localized to the plasma membrane where Fsk proteins are expected to be (There is no colocalization between HT-Fks1 and CSF-F647). The data suggest possible defects in either protein folding or transportation, although a fraction of HT-Fks1 can still confer resistance to CSF. Also, how is Fks2’s cellular localization affected in the fen1∆ mutant? These issues complicate the interpretation of the results, and the data are not sufficient to conclude that the observed reduced association of CSF-AF647 with the cell surface is responsible for the CSF resistance of the fen1∆ mutant. It is merely a correlation.

(3) Lines 265-270. SNPs in FKS1 HS1 did not affect the fitness of C. glabrata in the gut of mice treated with CSF for two weeks, while mutations in FKS2 increased the fitness. Does Fks1 and Fks2 respond differently to CSF? Why? A relevant question is why Fks2 cellular localization was not examined in Figure 3?

Reviewer #3: None

Reviewer #4: (No Response)

**Part III – Minor Issues: Editorial and Data Presentation Modifications**

Reviewer #1: N.A.

Reviewer #3: None

Reviewer #4: Minor comments:

Line 123, in summary rather than sum

Line 126 states intracellular levels of PHS, do you mean membrane levels?

Line 148, specify Candida species

Line 155 ergosterol

Line 209 Figure 3D

Figure 2A it is unconventional to use the term sterilization, you explain its meaning in the legend but as you are monitoring fungal counts in faeces you cannot be completely sure that no fungi remain in the GI tract. Why not state caspofungin treatment or plus caspofungin?

PLOS authors have the option to publish the peer review history of their article (what does this mean?). If published, this will include your full peer review and any attached files.

Reviewer #1: No

Reviewer #3: No

Reviewer #4: No
---

## [Decision Letter · Decision Letter 2]

19 Aug 2024

Dear Dr. Shor,

We are pleased to inform you that your manuscript 'Evolutionary dynamics in gut-colonizing Candida glabrata during caspofungin therapy: emergence of clinically important mutations in sphingolipid biosynthesis' has been provisionally accepted for publication in PLOS Pathogens.

Best regards,

James B. Konopka

Academic Editor

PLOS Pathogens

Michal Olszewski

Section Editor

PLOS Pathogens

Michael Malim

Editor-in-Chief

PLOS Pathogens

orcid.org/0000-0002-7699-2064

I am happy to say that the reviewers all agree that your manuscript makes an important contribution and is now acceptable for publication.

Reviewer Comments (if any, and for reference):

Reviewer's Responses to Questions

**Part I - Summary**

Reviewer #1: Please see my previous review.

Reviewer #3: My initial comments (Reviewer 3) still stand for this iteration of this manuscript.

In my opinion, the authors have been greatly responsive to reviewer critiques and have presented an improved manuscript. They have embraced most of the comments offered by reviewers 1 and 3, and where possible have undertaken additional experimentation as suggested.

Reviewer #4: Thank you for addressing comments of all reviewers. The revised manuscript has been strengthened by the changes made and other points are out of scope of this study as are major challenges in the field.

**Part II – Major Issues: Key Experiments Required for Acceptance**

Reviewer #1: The authors have conducted experiments in response to my comments, and the results and conclusions made are satisfactory.

Reviewer #3: None

Reviewer #4: Nothing further acquired.

**Part III – Minor Issues: Editorial and Data Presentation Modifications**

Reviewer #1: N.A.

Reviewer #3: None

Reviewer #4: None

PLOS authors have the option to publish the peer review history of their article (what does this mean?). If published, this will include your full peer review and any attached files.

Reviewer #1: No

Reviewer #3: No

Reviewer #4: No

---

## [Editor Report · Acceptance letter]

3 Sep 2024

Dear Dr. Shor,

We are delighted to inform you that your manuscript, "Evolutionary dynamics in gut-colonizing Candida glabrata during caspofungin therapy: emergence of clinically important mutations in sphingolipid biosynthesis," has been formally accepted for publication in PLOS Pathogens.

Best regards,

Michael Malim

Editor-in-Chief

PLOS Pathogens

orcid.org/0000-0002-7699-2064